# Thermally-induced atropisomerism promotes metal-organic cage construction

Jiaqi Liang [1], Shuai Lu[2], Yang Yang[1], Yun-Jia Shen[1], Jin-Ku Bai[1], Xin Sun[1], Xu-Lang Chen[3], Jie Cui[4], Ai-Jiao Guan[4], Jun-Feng Xiang [4], Xiaopeng Li [2], Heng Wang[2] ✉, Yu-Dong Yang[1,5] ✉ & Han-Yuan Gong [1] ✉

Molecular folding regulation with environmental stimuli is critical in living and artificial molecular machine systems. Herein, we described a macrocycle, cyclo[4] (1,3-(4,6-dimethyl)benzene)[4](1,3-(4,6-dimethyl)benzene)(4-pyridine). Under 298 K, it has three stable stiff atropisomers with names as **1** ($C_s$ symmetry), **2** ($C_s$ symmetry), and **3** ($C_{4v}$ symmetry). At 393 K, **1** can reversibly transform into **2**, but at 473 K, it can irrevocably transform into **3**. At 338 K, **3** and $(PhCN)_2PdCl_2$ complex to produce the metal-organic cage **4**. Only at 338 K does the combination of **1** or **2** and $(PhCN)_2PdCl_2$ create a gel-like structure. Heating both gels to 473 K transforms them into **4**. In addition to offering a thermally accelerated method for modifying self-assembled systems using macrocyclic building blocks, this study also has the potential to develop the nanoscale transformation material with a thermal response.

In both live organisms and artificial systems, macrocycles are crucial to the construction of self-assembling structures[1-6]. Responses to environmental stimuli are essential for changing these systems' particular architecture and associated features[7-12]. The stimuli may include light[7,13-16], pH value[17-19], guests (ion, solvents, small organic molecules, etc.)[1,8-10,19-26], ox/reduction[27], and/or temperature[2,3,12,28], etc. Functional macrocycles with stimulus responses are frequently used for catalysis[4,22,29], drug delivery[9,19,21,24], substance separation[4,24,30], adsorption[5,20], and recognition[9,10,21,31,32]. The temperature response in particular is a typical occurrence in biological systems. Additionally, it is used as a slipping technique in the production of rotaxane[17,27,33]. Some examples of thermal-induced conversion between two forms are achieved[28,34,35]. However, some biomacromolecules have been found to have more than two distinct stable or metastable folding configurations. Achieving heating-induced interconversion control between three or more different stiff isomers of artificial systems is still challenging[3,12,28,34].

Macrocycles have recently become a popular building element for creating metal−organic cages. The main examples included calixarenes[24,36-42], pillar[n]arene[43], and porphyrin[41,44-46], etc. These coordination cages are prospects for advanced functional materials due to their large interior volume, regulated cavity environment, and high porosity. Massive coordination cages and their potential applications have drawn a lot of attention, but corresponding approaches still have to be developed[47-51]. It is noteworthy that ligand modulation typically necessitates premodification, laborious synthesis, and/or guest addition in metal−organic self-assembly (such as metal−organic cages)[7,19,49,52-59]. Currently, macrocycles are often assembled in fixed forms. The development of controlling macrocycle structure and related assembly structure is still ongoing.

Herein, we describe a nanoscale transformation that uses thermal response to create an organic-metallic cage out of macrocyclic building blocks. The macrocycle is cyclo[4](1,3-(4,6-dimethyl)benzene)[4](1,3-(4,6-dimethyl) benzene)(4-pyridine). At 298 K, it possesses three distinct stable rigid atropisomers. Two are $C_s$ and one is $C_{4v}$ symmetry (named as **1**, **2**, or **3**, respectively). These three atropisomers are distinguishable from one another when heated at various temperatures. Especially, the heat transformation could be used to specifically produce atropisomers and to further control self-assembly stepwise or in situ. As an example, both approaches

[1]College of Chemistry, Beijing Normal University, Xinjiekouwaidajie 19, 100875 Beijing, P. R. China. [2]College of Chemistry and Environmental Engineering, Shenzhen University, 518060 Shenzhen, Guangdong, P. R. China. [3]College of Chemistry and Chemical Engineering, Hubei Key Laboratory of Pollutant Analysis and Reuse Technology, Hubei Normal University, 435002 Huangshi, P. R. China. [4]Institute of Chemistry Chinese Academy of Sciences, 100190 Beijing, P. R. China. [5]Department of Chemistry, The University of Texas at Austin, 105 East 24th Street, Stop A5300, Austin, TX 78712-1224, USA. ✉e-mail: hengwang@szu.edu.cn; yudongyang@utexas.edu; hanyuangong@bnu.edu.cn

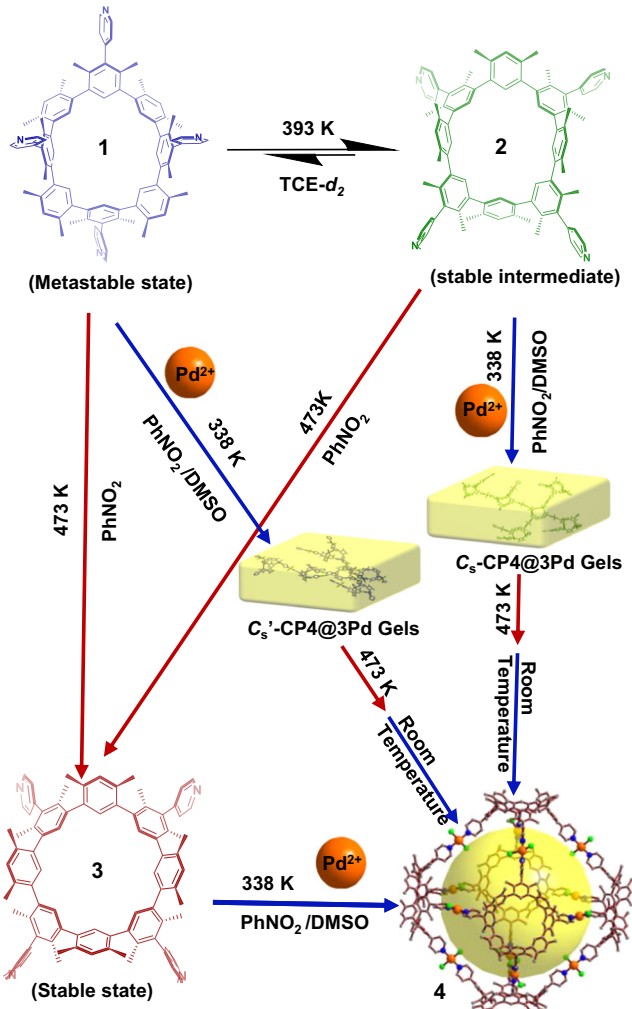

**Fig. 1 | The schematic diagram of heating regulation from macrocycle to further metal−organic cage 4 assembly in both stepwise and in situ paths.** **1**, and **2** coordinate with Pd²⁺ to form gels at temperature lower than 338 K, which can be converted into metal−organic molecular cages after heating at 473 K. The coordination between **3** and Pd²⁺ at 338 K can directly form the same cage **4**.

resulted in the generation of the octahedron-like metal−organic cage **4** (Fig. 1). Yellow gels are formed in situ when **1** or **2** and Pd²⁺ are combined. The macrocyclic rigid backbone flips when the gel is heated. The initial gels are further altered to produce an autonomous cage structure. To the best of our knowledge, this is a singular case of a thermally responsive macrocycle to self-assembly transition at the nanoscale. This method offers a fresh approach to thermally responsive material preparation.

## Results

### Synthesis and structural characterization of macrocycle

Macrocycle was generated as shown in Fig. 2. Under acidic conditions, 1,5-dibromo-3-iodo-2,4-dimethylbenzene (**5**) is generated from 1,5-dibromo-2,4-dimethylbenzene with a yield as 68%. 4-(2,6-dimethyl-3,5-bis(4,4,5,5-tetramethyl-1,3,2-dioxaborolan-2-yl)phenyl) pyridine (**7**) was obtained from **5** using Suzuki−Miyaura coupling and following boron esterification with two-step overall yield as 64%. Compound **8** was created from 1,5-dibromo-2,4-dimethylbenzene in 95% yield. Then compound 4-(5,5″-dibromo-2,2″,4,4″,4″,6′-hexamethyl-[1,1′,3′,1″-terphenyl]−5′-yl)pyridine (**9**) was produced via Suzuki−Miyaura coupling reaction between **7** and **8** with yield as 79%. Finally, the Suzuki−Miyaura coupling cyclization between **7** and **9** gave out the target macrocycle

cyclo[4](1,3-(4,6-dimethyl)benzene)[4](1,3-(4,6-dimethyl)benzene)(4-pyridine).

Three stable isomers of macrocycle were discovered to exist, and they could all be successfully separated using a silica gel column. In high-resolution matrix-assisted laser desorption/ionization time of flight (MADLI-TOF) mass spectrometry, these isomers exhibit the identical molecular weight values (Supplementary Figs. 28–30). But in the traveling-wave ion mobility mass spectra (TWIM-MS), three isomers had separate drift durations (4.78 ms, 4.57 ms, or 4.87 ms, respectively corresponding to **1, 2**, or **3**) (Supplementary Figs. 31–33, collision voltage: 0 V)[49,60]. A higher temperature (473 K) effectively caused the change from **1** and **2** to **3** in the gas phase. More research (see *infra*) revealed that the yield of the stiff atropisomer with $C_s$ or $C_{4v}$ symmetry (i.e., **1, 2**, or **3**) was 3.2, 6.1, or 1.5%, respectively (Fig. 2). Theoretical calculations were employed to gain an in-depth understanding of the various cyclization yields of these atropisomers (using molecular mechanics (MM + ) or semiempirical techniques (PM7) method) (Supplementary Fig. 34). With a Pd catalyst linker, their corresponding cyclization intermediates were valued. It is discovered that the formation heat of the **2** or **1** associated intermediates (i.e., **2-im** or **1-im**) is lower than that of **3** related intermediates (i.e., **3-im**) (Supplementary Table 1). However, later research revealed that at temperatures greater than 348 K, **1** can be transformed into **2**. These findings suggested that at least both of the two factors, namely intermediate stability and atropisomer conversion, should be considered to determine the yield distribution of final products.

Three atropisomers were further confirmed via single-crystal X-ray diffraction analysis of **1**•2.25CH₂Cl₂, **2**•CH₂Cl₂•0.75H₂O and **3**•CH₃COOC₂H₅•2CH₃CN•3.5H₂O (Fig. 3b–d, Supplementary Figs. 35–40, and Supplementary Table 2). Each single crystal example was grown with slow evaporation from the relevant isomer solution (1 mM). It should be noted that the corresponding macrocycle isomer has the structure that is similar to meso-substituted resorcinarene with chair (**1**), kite (**2**), or crown (**3**) comformation[35]. With the NMR spectroscopic study, the similarities were further verified. Four 4-(2, 6-dimethylphenyl) pyridinyl (**P**) and four *m*-xylene (**B**) fragments alternately construct each macrocycle. In the structure of **1**•2.25CH₂Cl₂, **1** has an elliptical cavity with major and minor distances of 8.6 and 7.7 Å, respectively (Fig. 3a and Supplementary Fig. 35). Its symmetry plane passes through opposite asymmetric **P** moieties (Fig. 3a). The other two symmetric **P** moieties are parallel to each another and pointing towards the same direction of the macrocycle. Two symmetrical **B** group sets point to the opposite direction. Each symmetric pyridinyl has an angle of 64° or 173° with its neighboring asymmetric pyridinyl group. The torsion angles between the adjacent aromatics are between 117.0(2)° and 119.2(2)° (Supplementary Fig. 36). With the location of adjacent **B** and **P** fragments in a sequence switched, **2** has the same cavity as **1** in the structure of **1**•2.25CH₂Cl₂ (Fig. 3b). A symmetry plane crosses two opposing **B** moieties. There is an angle of 85° or 90° between symmetrical pyridinyl groups, respectively. The neighboring asymmetrical pyridinyls are at an angle of 57° (Fig. 3b). The torsion angles between neighboring **B** and **P** range from 117.1(1)° to 190.7(5)° (Supplementary Fig. 38). In **3**•CH₃COOC₂H₅•2CH₃CN•3.5H₂O, four plumb symmetry planes pass through the macrocycle's $C_4$ axis (Fig. 3c). A double-bowl-shaped cavity is created by four **P** groups and four **B** groups. The cavity depth of **3** was approximately 8.4 Å, and the upper and lower rims of the structure had approximate diameters of 12 Å and 8.6 Å (Fig. 3c and Supplementary Fig. 39). Four **P** groups with respective pyramidal angles of 40°, 50°, 61°, and 66° point to the same side of the macrocycle. The four **B** units can form another pyramid when they all point to a different side of the macrocycle. In this case, the torsion angle between the adjacent planes is between 117.5(4)° and 121.2(8)° degrees (Supplementary Fig. 40).

The solution NMR spectroscopic study is carried out in tetrachloroethane-*d₂* (TCE-*d₂*) at 298 K (Fig. 4 and Supplementary Figs. 11–22). The ¹H NMR spectra of each rigid **1-3** atropisomer (Fig. 4b)

**Fig. 2 | Synthesis of macrocycle.** All yields refer to column chromatography purified products. More synthetic details are shown in "Methods".

exhibits a single set of high-resolution signals. The $C_s$ symmetry of fixed **1** or **2** is still adopted in solution, according to four independent pyridine proton signals. Their [13]C NMR spectra, each of which had 39 or 38 carbon signals, provided additional evidence in support of the deductions (Supplementary Figs. 12 and 14). The significantly less complex signals that were seen in the [1]H (2 independent pyridine proton signal) and [13]C NMR (15 carbon signals) spectra of **3** suggest that the structure has higher symmetry in solution (Fig. 4b and Supplementary Fig. 16).

**Heat driven conversion between various atropisomers**
There are many natural compounds that exhibit biaryl atropisomerism. It has a significant impact on their bioactivities. As typical species of biaryl moieties, the biphenyl group was introduced during the construction of small chemical compounds, macrocycles, etc[3]. Atropisomers, including diastereoisomers and enantiomers, are produced as a result. There are still issues with the complicated atropisomer distribution and conversion between more than two rigid atropisomers[28,61]. The related research may give us a greater understanding of the specific atropisomer production, interconversion, and related features in living systems.

In our initial work, **1** or **2** was loaded on either a Celite or a NaCl substrate before being heated at 573 K in an argon environment for one hour. **1** or **2** was quantitatively converted to **3** according to thin layer chromatography (TLC) and reaction product analysis (Supplementary Fig. 41).

Further atropisomer transformation study in solution was carried out. Since PhNO$_2$-$d_5$ is the commercially available deuterated solvent with the greatest boiling point (483 K), the temperature-dependent [1]H NMR spectra of **1**, **2**, or **3** were collected from 298 to 473 K. Heating caused most **1** to convert to **2** at 348–418 K (Supplementary Fig. 42). All **3** proton signals manifest in either the case of **1** or **2** at temperatures higher than 418 K. **1** or **2** completely transformed into **3** at 473 K (Supplementary Figs. 42 and 43).

Time-dependent [1]H NMR detection of each atropisomer was performed at 393 K (Fig. 5 and Supplementary Figs. 44–50) or 473 K

(Fig. 6 and Supplementary Figs. 51–53), respectively, to better understand the heating-induced conversion between **1**, **2**, and/or **3**.

At 393 K, the **1** spectra in PhNO$_2$-$d_5$ exhibit gradually decreasing signals and produce **2** signals with increasing intensity ratios until the balance is reached after 6 minutes (Fig. 5b and Supplementary Fig. 44). The ratio of **2** to **1** was thereafter maintained at 86:14 (Fig. 5c and Supplementary Fig. 45a). The system reaches dynamic equilibrium with a similar ratio between **2** and **1** as 85:15 after heating **2** solution for 1 min at 393 K (Supplementary Figs. 46 and 47). It should be highlighted that the reversible conversion between **1** and **2** occurred at 393 K without the involvement of **3** (Fig. 5a–c and Supplementary Figs. 44 and 46). One explanation for this could be that the temperature is too low to allow **1** or **2** to pass through the energy barrier and create **3**. At 393 K, the change from **1** to **2** was considered to be a first-order reversible process. The thermal dynamics and kinetic parameters of the transition were calculated using time-dependent [1]H NMR data in accordance with the standard equilibrium (Eq. 1). These parametes include the reaction rate constants ($k_1 = (1.4 \pm 0.1) \times 10^{-2}$ s$^{-1}$, $k_{-1} = -(2.3 \pm 0.1) \times 10^{3}$ s$^{-1}$) (Supplementary Fig. 45b), the Gibbs activated free energy ($\Delta G^{\ddagger}_{I(393\,K)}$ as 111 ± 6 kJ mol$^{-1}$), the equilibrium constant ($K_1$ as 6.1 ± 0.2), and the Gibbs free energy ($\Delta G^{\theta}_{I(393\,K)}$ as −5.9 ± 0.2 kJ mol$^{-1}$) (Supplementary Table 3). The energy map of the transition between **1** and **2** in PhNO$_2$-$d_5$ at 393 K can be found in Fig. 5d using the formation energy of **1** as standard 0 kJ mol$^{-1}$. Also performed at 393 K were the time-dependent [1]H NMR spectra of **1** or **2** in TCE-$d_2$ solution (Supplementary Figs. 48 and 50). After 8 min, the equilibrium of **2** as the substrate is stable, and the ratio of **2** to **1** is maintained at 79:21 (Supplementary Fig. 49a). Dynamic equilibrium was achieved with a ratio of 80:20 between **2** and **1** after heating **2** in TCE-$d_2$ for 1 min at 393 K (Supplementary Fig. 49d). The thermal dynamics and kinetic parameters of the transition from **1** to **2** in TCE-$d_2$ at 393 K were determined using the same standard equilibrium (Eq. 1) and data analysis method. These parameters involve the reaction rate constants ($k_1$ as $(5.2 \pm 0.4) \times 10^{-3}$ s$^{-1}$, $k_{-1}$ as $-(1.4 \pm 0.1) \times 10^{3}$ s$^{-1}$) (Supplementary Fig. 49b), the Gibbs activated free energy ($\Delta G^{\ddagger}_{I(393\,K)}$ as 114 ± 6 kJ mol$^{-1}$,

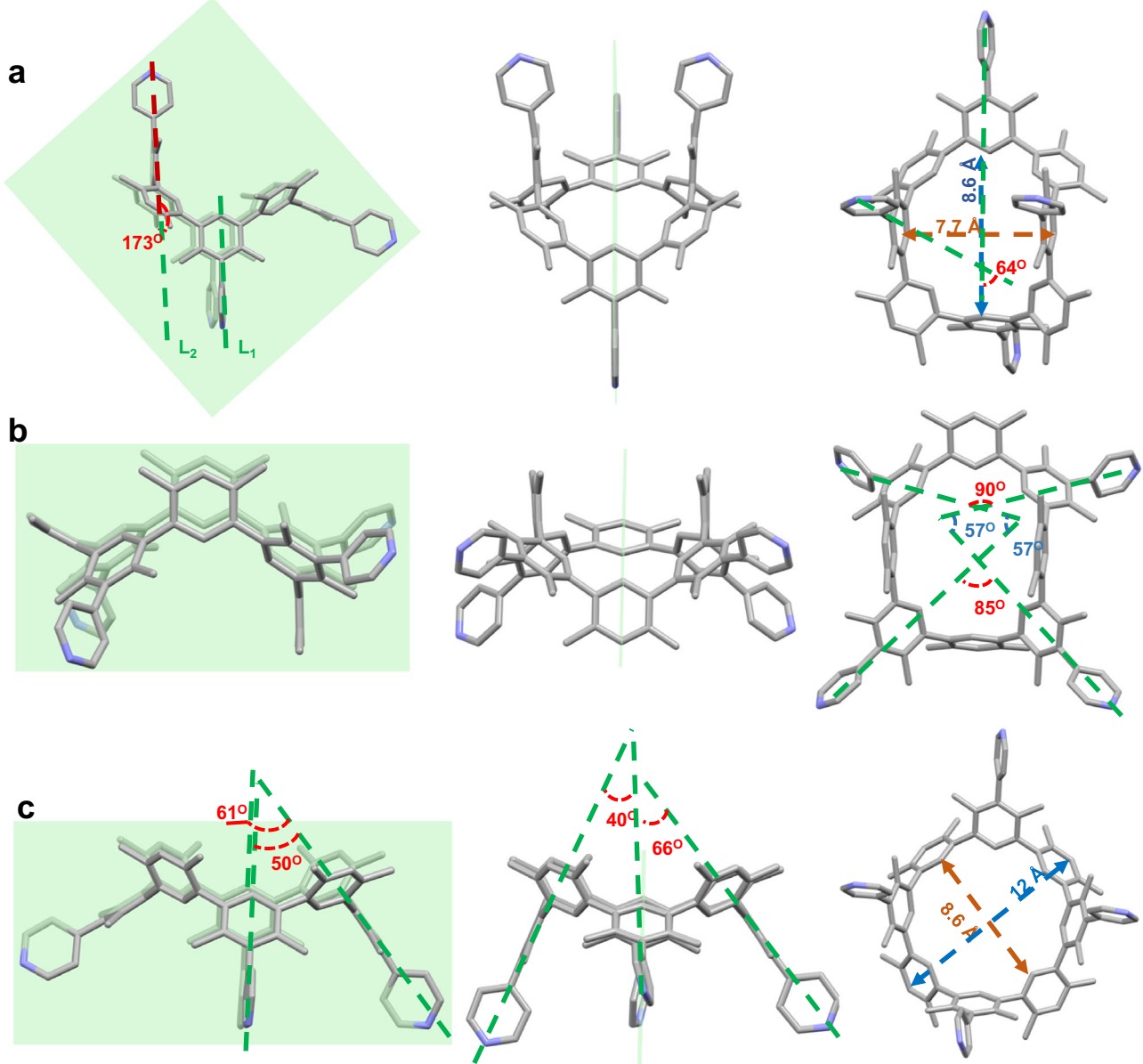

**Fig. 3 | Single crystal X-ray diffraction structures of macrocycles (1, 2, and 3).** The associated angle between the **P** and **B** moieties and the cavity of **1** (**a**), **2** (**b**), and **3** (**c**) hydrogen atoms and solvent molecules are omitted for clarity. The yellowish green plane is the molecular symmetry plane. **a** uses the geometric mathematics technique of parallel lines to illustrate the angle between two **P** groups (dashed line $L_1$ is parallel to dashed line $L_2$). Carbon: silvery gray, Nitrogen: bluish violet.

the equilibrium constant ($K_1$ as 3.7 ± 0.2), and the Gibbs free energy ($\Delta G^{\theta}_{1(393\,K)}$ as −4.4 ± 0.2 kJ mol$^{-1}$) (Supplementary Fig. 49c). When **1** to **2** conversion parameters are compared between the cases in PhNO$_2$-$d_5$ and TCE-$d_2$ at 393 K, it is discovered that the transition in PhNO$_2$-$d_5$ has a higher reaction rate ($k_1$ as (1.4 ± 0.1) × 10$^{-2}$s$^{-1}$) than that in TCE-$d_2$ ($k_1$ = ((5.2 ± 0.4) × 10$^{-3}$ s$^{-1}$). In both solvents, there is just a comparable reversible conversion process (without **3** participation). It is inferred that the solvent can influence the same isomerization process at the same temperature, including its kinetics and thermodynamics, but it cannot drive reactions with greater energy barriers, such as the conversion of **1** or **2** to **3**.

$$
\begin{array}{c}
k_1 \\
\mathbf{1} \rightleftharpoons \mathbf{2} \\
k_{-1} \\
K_1 = k_1/k_{-1}
\end{array}
\qquad (1)
$$

In PhNO$_2$-$d_5$ at 473 K, a further time-dependent $^1$H NMR spectroscopic investigation of **1, 2,** or **3** was carried out (Fig. 6a, b and Supplementary Figs. 51 and 53). Most of the **1** (approximately 82.3%) swiftly changed to the main product **2** (63.8%) and the minor product **3** (18.5%) after heating at 473 K for 2 min (Fig. 6c and Supplementary Fig. 52a). Then, as **3** increased, the molar ratio of **2** steadily decreased. The ratio of **1, 2,** or **3** reached equilibrium at 1.00:0.3:98.7 after 100 min. It was discovered in the previous work that the final equilibrium ratio between **2** and **1** was around 6:1 at 393 K. The ratio of **2** to **1** changes to 3.50:1 at 473 K (Fig. 6c). The equilibrium at 473 K showed that nearly all of the **1** and **2** signals disappeared. The thermal change from **1** or **2** to **3** at 100 min is nearly quantitative (Supplementary Figs. 51 and 53). According to the data displayed above, the overall process at 473 K included a quick transformation from **1** to **2** and a slow transition from **2** to **3** (Fig. 6a). Here, the metastable atropisomer **1** underwent conversion to the stable state **3**, with **2** serving as a stable intermediate. It is possible to convert **1** to **3** in stages or all at once

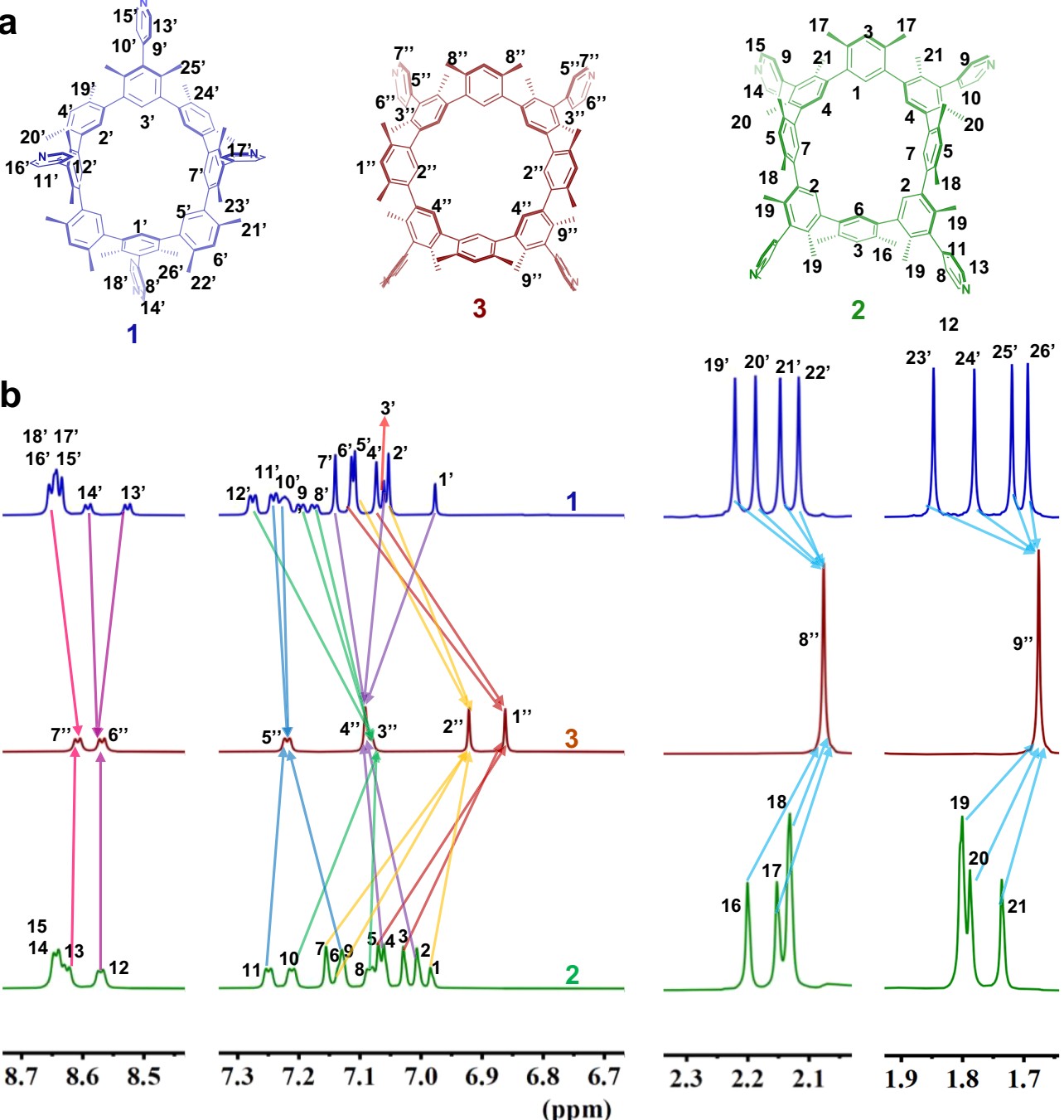

**Fig. 4 | Structures and ¹H NMR spectroscopic characterizations of artopisomers.** The structures (**a**) and corresponding ¹H NMR spectra (**b**) of **1** (up), **3** (middle), and **2** (bottom) (each concentration as $1 \times 10^{-2}$ M in TCE-$d_2$ at 298 K; 500 MHz).

under high temperatures. Here, rigid macrocycle structures are regulated in an orderly manner.

The time-dependent ¹H NMR data at 473 K were used to compute the thermodynamic and kinetic parameters of both processes in the conformer transition in accordance with the conventional equilibrium (Eq. 2). The calculation results in reaction rate constants ($k_1$ as $(8.0 \pm 0.4) \times 10^{-2}$ s⁻¹; $k_{-1}$ as $-(2.3 \pm 0.1) \times 10^{-2}$ s⁻¹; $k_2$ as $(2.1 \pm 0.1) \times 10^{-3}$ s⁻¹; $k_{-2}$ as $-(7.9 \pm 0.1) \times 10^{-5}$ s⁻¹) (Supplementary Fig. 52b and Supplementary Table 4), the Gibbs activated free energy values ($\Delta G^{\neq}_{1(473\,K)}$ as $128 \pm 6$ kJ mol⁻¹; $\Delta G^{\neq}_{2(473\,K)}$ as $142 \pm$ kJ mol⁻¹) (Supplementary Table 4), the equilibrium constants ($K_1$ as $3.5 \pm 0.2$; $K_2$ as $27 \pm 1$), and the Gibbs free energy values ($\Delta G^{\theta}_{1(473\,K)}$ as $-4.3 \pm 0.3$ kJ mol⁻¹; $\Delta G^{\theta}_{2(473\,K)}$ as $-13 \pm 1$ kJ mol⁻¹) (Supplementary Table 4). The potential energy diagram of the

conversion between three atropisomers at 473 K can be produced via taking into account the formation energy of **1** as standard 0 kJ mol⁻¹ (Fig. 6d). To comprehend the processes of transformation from **1** to **2** and finally to **3**, theoretic calculations (using the SemiEmpirical PM7 method) were performed (Supplementary Fig. 54 and Supplementary Table 5). The computed values are in line with the trend of the experiment results (Supplementary Figs. 52 and 54 and Supplementary Tables 4 and 5).

$$
\begin{array}{ccc}
k_1 & & k_2 \\
\mathbf{1} \rightleftharpoons & \mathbf{2} & \rightleftharpoons \mathbf{3} \\
k_{-1} & & k_{-2}
\end{array}
$$
$$
K_1 = k_1/k_{-1}, K_2 = k_2/k_{-2}
$$

(2)

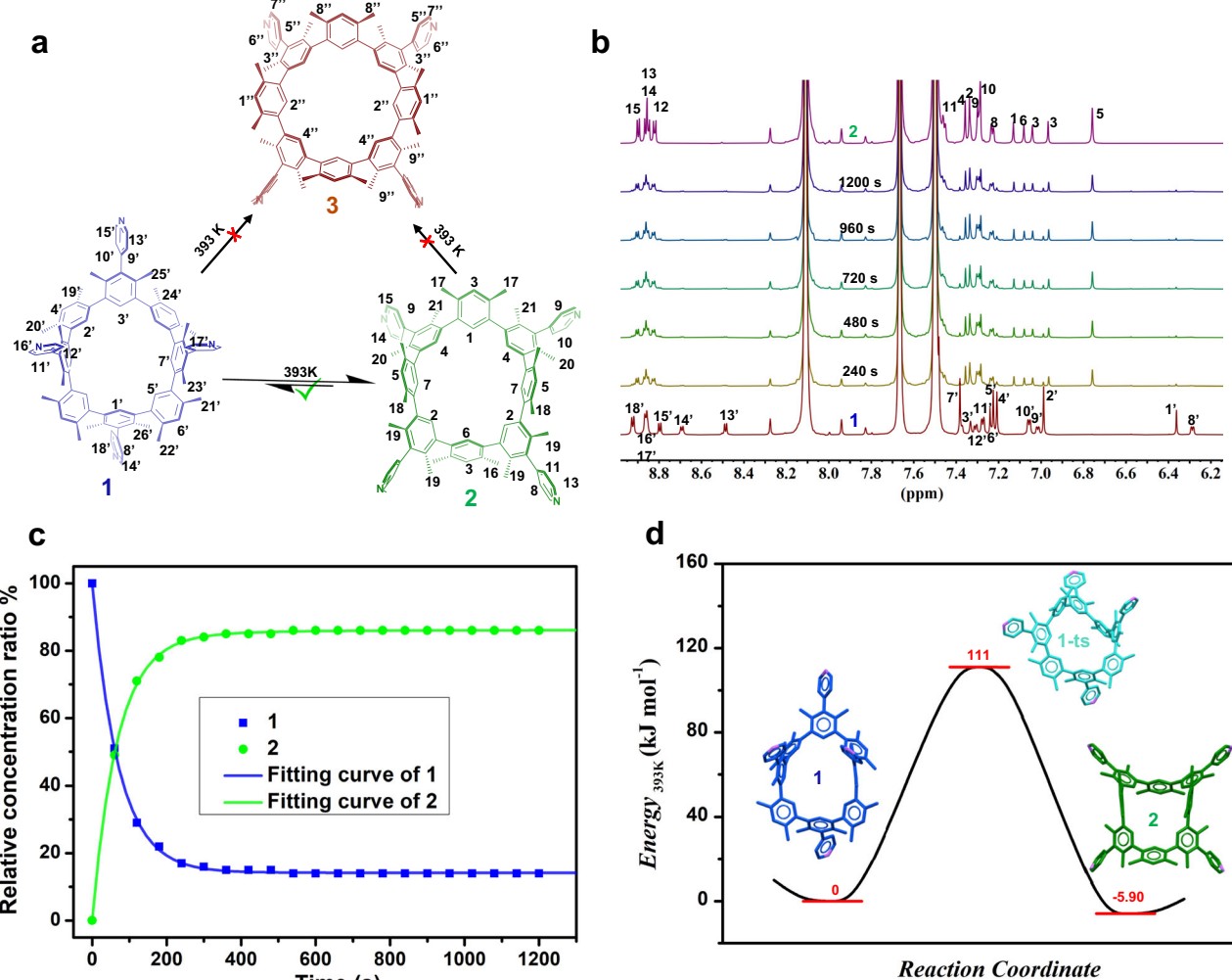

**Fig. 5 | The reversible transformation between rigid 1 and 2 at 393 K.**
**a** Schematic representation of the conversion between **1**, **2** and **3** in PhNO$_2$-$d_5$ at 393 K. **b** Time-dependent $^1$H NMR spectra of **1** (5.3 × 10$^{-3}$ M) in PhNO$_2$-$d_5$ at 393 K. **c** Time-dependent relative concentration ratio changes of **1** (blue dot) and **2** (green dot) in PhNO$_2$-$d_5$ at 393 K. **d** the potential energy diagram of the conversion between **1** and **2** in PhNO$_2$-$d_5$ at 393 K. The transition state is listed based on theoretical calculation.

Further atropisomer transition study was carried out in the gas phase with TWIM-MS at various collision voltages. As the collision energy increased, **1** changed to **2** (Supplementary Fig. 31), and then **1** changed to **3** (Supplementary Fig. 32). In the case of **3**, however, no atropisomer transition was seen. These findings, which are in excellent agreement with those of solid and solution experiments, further support the hypothesis that **3** is the energetically preferred isomer. It should be emphasized that despite being subjected to high collision energies (150 V), there was no discernible breakup of these macrocycles. These results demonstrated their extraordinary gas phase stability.

The introduction of pyridine groups on cyclo[4](1,3-(4,6-dimethyl)benzene)[4](1,3-(4,6-dimethyl)benzene)(4-pyridine) increases the number of stable atropisomer types to three at room temperature. It is opposed to the previously reported cyclo[8](1,3-(4,6-dimethyl)benzene) (**CDMB-8**) involving two rigid atropisomers[3]. A higher temperature (473 K) triggered an irreversible transformation from metastable forms to stable conformation, as seen in **1-3** system, while a lower temperature (393 K) caused reversible transformation between metastable atroisomers. The **CDMB-8** system does not display this outcome. Additionally, rigid atropisomer conversion involving porphyrin[62], [4]cyclo-chrysene[12], or [n]cyclo-4,10-pyrenylenes[63], among others, required a temperature at least 473 K. In the **1-3** case, fast conversion

between the metastable atropisomer requires a much lower temperature (393 K). Additionally, in comparison to **CDMB-8**, the complete molecular skeleton and the nearby methyl groups around the flip bond involved in the conformational transformation remain unchanged. It is important to note that the addition of long-range groups (e.g., pyridinyl) causes **1-3** and **CDMB-8** to differ in terms of isomer type, transition temperature, and transition mechanism.

In contrast to other macrocyclic compounds that have currently undergone significant study (such as calixarene, porphyrin, or cyclic arene), chemical modifications are typically needed to incorporate various substituents for conformation fixing[15,63,64]. In these systems, comformer regulation requires the participation of the guest[10,11,15], pH regulation[18], high temperature[62], etc. The regulation method of macrocyclic structures is supplemented with the example involving **1-3**, which solely uses heat to control transformation between ambient temperature stable conformers.

## Construction and single crystal X-ray diffraction analysis of metal-organic cage 4

Metal-organic self-assemblies, such as metal-organic cages, are frequently built using the pyridinyl group as a powerful ligand[6,41,64-66]. It was believed that using macrocycles with pyridinyl groups as their building blocks was an efficient way to create specific topological

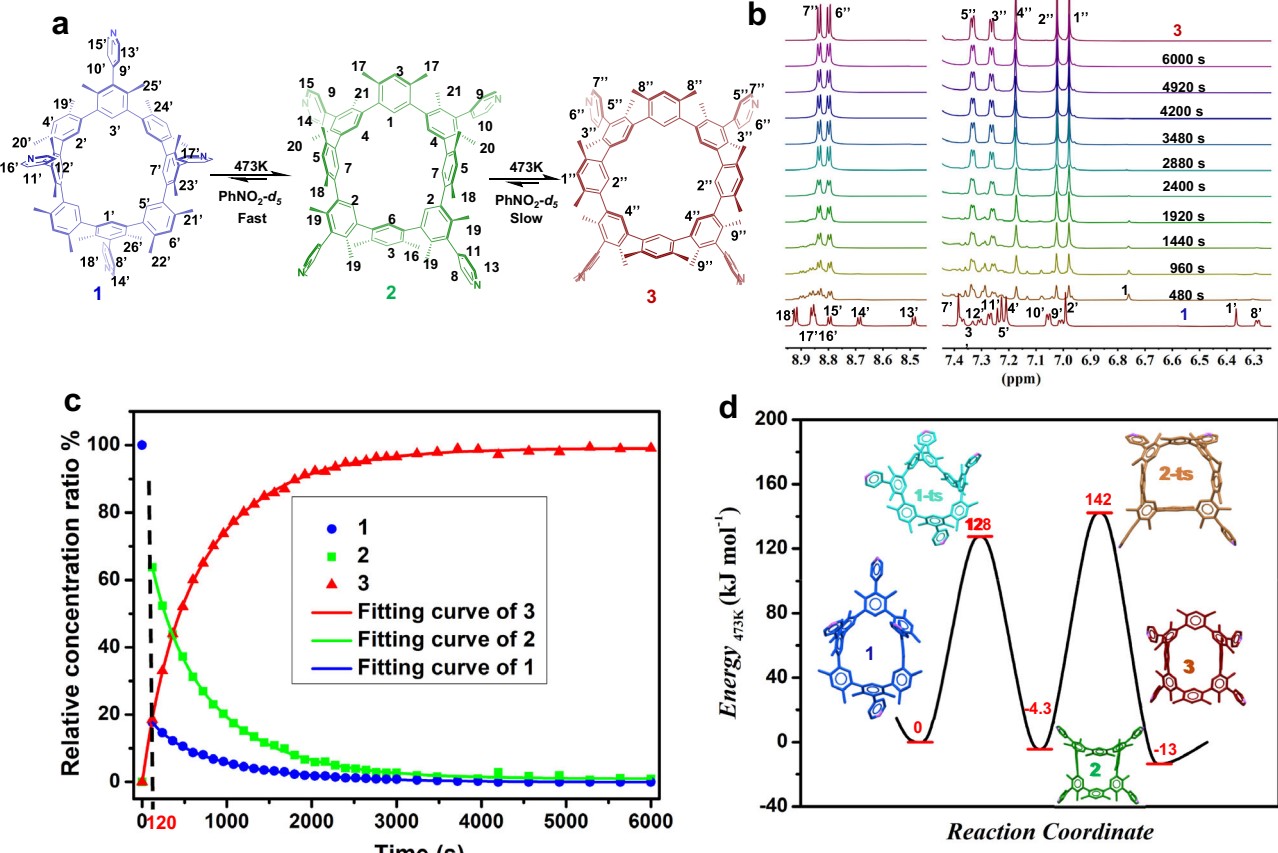

**Fig. 6 | The transformation between three atropisomers of 1–3 at 473 K.**
**a** Schematic representation of the transformation between **1**, **2**, and **3** in PhNO$_2$-$d_5$ at 473 K. **b** Time-dependent $^1$H NMR spectra of **1** (5.3 × 10$^{-3}$ M) in PhNO$_2$-$d_5$ at 473 K. **c** Time-dependent conversion between **1** (blue dot), **2** (green dot), and **3** (red dot) and related non-linear fitting curves (solid lines) at 473 K. **d** The potential energy diagram for the conversion between **1**, **2**, and **3** in PhNO$_2$-$d_5$ at 473 K. The transition states are listed based on theoretical calculation.

structures with large cavities. Herein, the self-assembled metal–organic cage **4** was generated. **3** (0.2 mg ml$^{-1}$) and (PhCN)$_2$PdCl$_2$ with a molar ratio as 1:2 were mixed in THF/DMF (1/4), THF/DMSO (1/4), or TCE/DMF (1/4) at 338 K for 12 h. The same metal–organic cage is generated in each case. The crystal has a large unit cell size, according to a single crystal X-ray diffraction analysis (*a* = 30.115(4) Å, *b* = 30.115(4) Å, *c* = 53.271(11) Å, *α* = 90.00°, *β* = 90.00°, *γ* = 90.00°, *V* = 48313(17) Å$^3$) (Supplementary Table 6). A massive octahedral structure can be seen in each individual cage (Fig. 7a and Supplementary Fig. 55). It contains two **10** (twisted **3** with $C_{2v}$ symmetry; the yellow macrocycles at the top as shown in Fig. 7d) and four **3** (the red macrocycles in the middle as shown in Fig. 7d, e). Each Pd$^{2+}$ in this case acted as a linker between two pyridinyl groups. Through the coordination of pyridinyl and Pd$^{2+}$ at an angle of 180.0°, each **10** binds to all four **3** (Fig. 7d). The remaining pyridinyl groups in the middle **3** coordinate with Pd$^{2+}$ at an angle of 175.8° or 178.2° to form a circle (Fig. 7e). The octahedron produced with the pyridine groups on **10** has top pyramidal angles of 44° (Fig. 7d). The middle pyramidal angles of the octahedron formed between the pyridine groups on **3** are 42.5°, and 54.3° (Fig. 7e). The enormous ellipsoidal cavity and high porosity of **4** are notable (Fig. 7e). The greatest and minimum diameters are 43 and 29 Å, respectively.

The stability of **4** in solution was examined using $^1$H NMR and $^1$H DOSY spectra in THF-$d_8$/DMSO-$d_6$. Heating the mixture containing **3** with 2 molar equiv. of (PhCN)$_2$PdCl$_2$ in THF-$d_8$/DMSO-$d_6$ at 338 K for 12 h, it produced a complex-related $^1$H NMR spectrum. The limited solubility of the assembly, coordination-restricted cyclo[4](1,3-(4,6-dimethyl)benzene)[4](1,3-(4,6-dimethyl)benzene)(4-pyridine) molecular mobility, and a putative exchange between **10** and **3** were the

specific causes of the $^1$H NMR spectrum complexity. The PyH$_{6'',7''}$ signal splitting disorder, low field shift, and disappearance of the PyH$_{4'', 5''}$ signal were all caused with increasing Pd$^{2+}$. These results validated the complexation between between **3** and Pd$^{2+}$ (Fig. 7b). $^1$H DOSY NMR further indicated the single product formation with only one diffusion coefficient (D) as 7.47 × 10$^{-11}$ m$^2$ s$^{-1}$ in THF-$d_8$/DMSO-$d_6$ (1/4) at 298 K (Fig. 7c). Furthermore, the radius of the complex $r_H$ as 2.0 nm can be calculated[47]. It is close to the **4** radius shown in the single crystal structure. Due to the restricted solubility, the $^{13}$C NMR data analysis of **4** was unsuccessful. The inability to obtain satisfactory mass spectrometry data of **4** was caused by the macrocycle poor ionization and the **4** limited solubility. **4** was synthesized on a larger scale. With a yield of 75%, **3** (20.0 mg) generated a 24.5 mg **4** sample (see Supplementary Fig. 56). According to the PXRD analysis, the larger-scale sample's pattern is similar to the simulated pattern from the **4** single crystal structure, indicating that the sample's structure is consistent with that of the **4** single crystal.

**Thermal promoted construction of metal–organic cage**
The precursor of **4**, **3** was synthesized with a low yield of only 1%. Thermal conversion of **2** or **1** could result in **3**, which is then assembled to form **4**. The progressive process, however, is time and labor-consuming. In order to create the cage **4** in situ, efforts were attempted to directly convert the complex between **1** or **2** and Pd$^{2+}$. In all of the investigated cases at 338 K, the mixture of **1** or **2** and Pd$^{2+}$ only generated yellow gels and was unable to obtain crystals (Fig. 8). The inconsistent orientation mismatch of the pyridinyl groups on the macrocycle for the construction of a fixed cage or mesh assembly

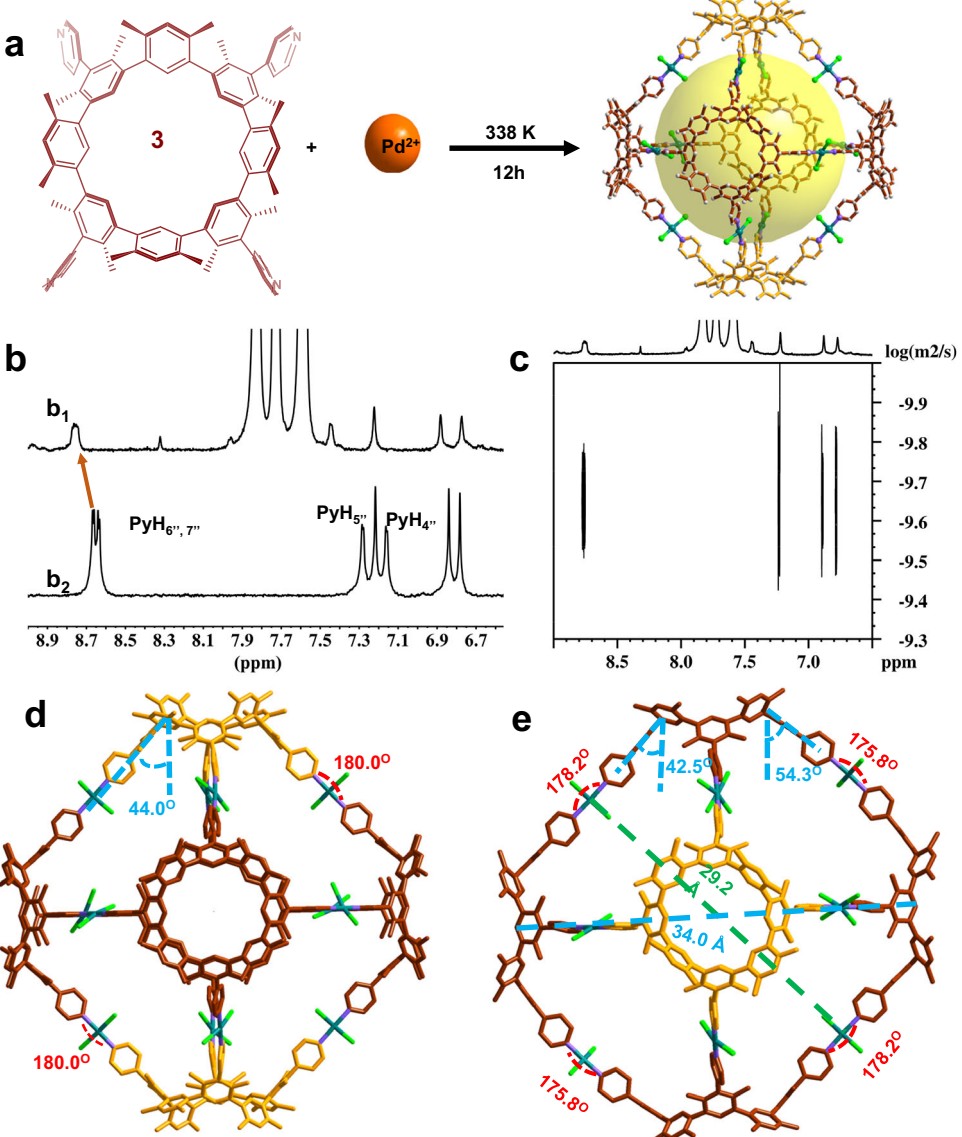

**Fig. 7 | The assembly process and single crystal X-ray structural analysis of 4.** **a** The scheme presentation of the complexation between **3** and Pd$^{2+}$ ion with a 1:2 molar ratio in THF/DMSO at 338 K for 12 h. **b** $^1$H NMR spectrum of **3** in the presence (b$_1$) and absence (b$_2$) of 2 molar equiv. of (PhCN)$_2$PdCl$_2$. **c** $^1$H DOSY spectrum of the complexes containing **3** and Pd$^{2+}$ (600 MHz, THF-$d_8$/DMSO-$d_6$ (1/4), 298 K). Single crystal X-ray diffraction structures of **4** (**d**, **e**, top and side view, respectively). Color code: C red or yellow, N purple, Pd bluish green, Cl green. Hydrogen atoms and solvent molecules have been omitted for clarity. The yellow macrocycle is $C_{2v}$ symmetric, and the red macrocycle is $C_{4v}$ symmetric.

could be the reason. When the molar ratio of Pd$^{2+}$ to macrocycle was not less than 2, the proton signals of **1** or **2** entirely disappeared in $^1$H NMR titration tests (Supplementary Figs. 57 and 58). Scanning electron microscopy (SEM) displayed all gels (i.e., **1**@2 Pd or **2**@2 Pd, respectively) were formed as nanospheres or plates. Numerous micron-sized pores of the microstructure can be found (Supplementary Fig. 59). According to the dynamic light scattering (DLS) test, the particle sizes of **1**@2 Pd or **2**@2 Pd are almost identical and significantly larger than those of **3**@2 Pd (Supplementary Fig. 60). The gel permeation chromatography (GPC) results implied the polymer produced between **1** or **2** and Pd$^{2+}$ has an average molecular weight that is roughly 20 times of the comparable macrocycle (Supplementary Figs. S61 and S62). All of these findings indicated that **1** or **2** and Pd$^{2+}$ forms polymeric structures. The calculation (with molecular mechanics (MM + ) method) implied that each atropisomer is easy to form infinite structures with Pd$^{2+}$ (Supplementary Figs. 63–67). At high temperatures, only **3** finally generates a distinct structure (Supplementary Table 7). It has been

proposed that the cage building process depends on the entropy effect of the dissociated ligands and solvents.

The complex of **3** and Pd$^{2+}$ has been found to have nearly identical $^1$H NMR, $^1$H DOSY, and theoretical radius values calculated with diffusion coefficient in THF-$d_8$/DMSO-$d_6$ or PhNO$_2$-$d_5$/DMSO-$d_6$ systems (Supplementary Figs. 68–70). On the basis of this fact, we attempt to synthesize the cage in situ using a PhNO$_2$-$d_5$/DMSO-$d_6$ solvent. The amount of Pd$^{2+}$ in the reaction was increased to 3 molar equiv because some of the Pd$^{2+}$ changed to palladium black at high temperature (473 K). When Pd$^{2+}$ and **1** or **2** are mixed at 338 K, a yellow gel is created immediately. (i.e., **1**@3 Pd or **2**@3 Pd, respectively). The proton signals of $^1$H NMR spectra changed similarly to **4** after heating **1**@3 Pd or **2**@3 Pd at 473 K for 1 h (Fig. 8, Supplementary Fig. 71). Additionally,$^1$H DOSY NMR showed single product generation with a diffusion coefficient. The diffusion coefficient of **1**@3 Pd or **2**@3 Pd is 7.21 × 10$^{-11}$ m$^2$ s$^{-1}$ or 5.96 × 10$^{-11}$ m$^2$ s$^{-1}$, corresponding to $r_H$ as 1.6 nm or 1.9 nm, respectively (Supplementary Figs. 72 and 73)[47]. Both $r_H$ values are in good

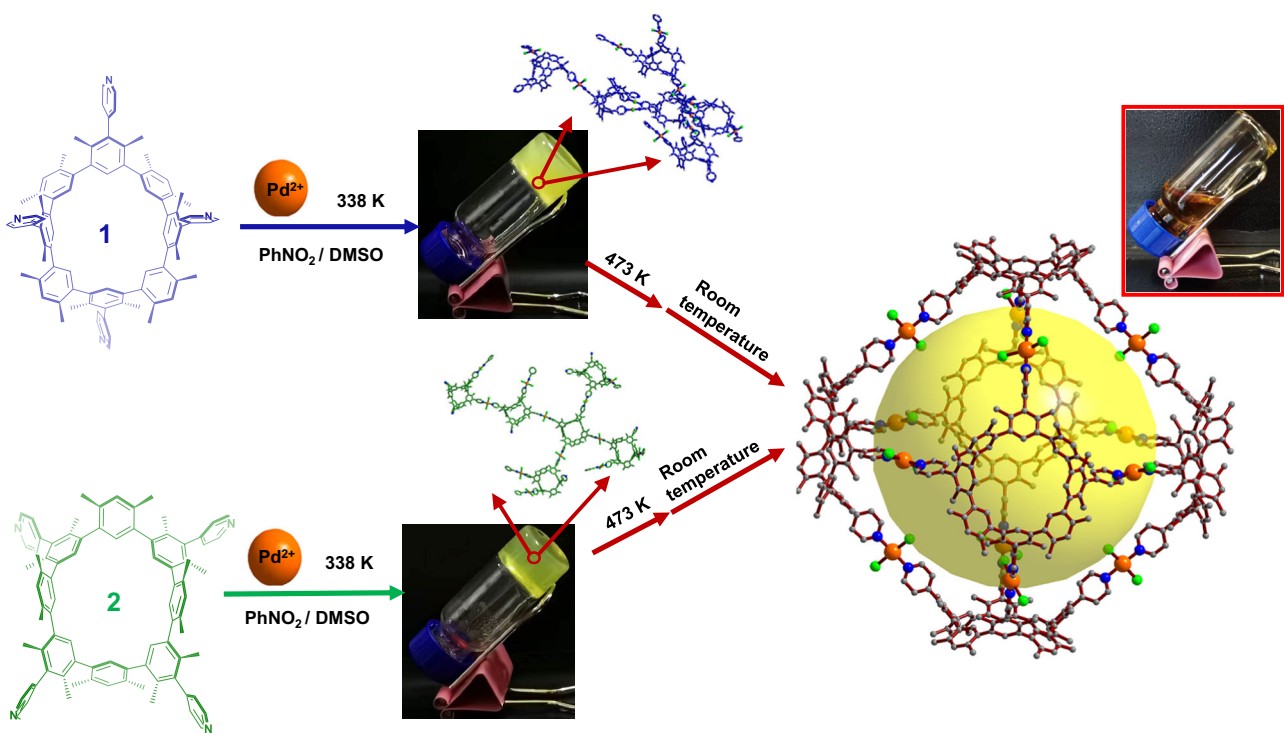

**Fig. 8 | The cage 4 construction from the mixture containing 1 or 2 and Pd²⁺ in situ.** The magnified image in the middle is the optimal structure obtained by calculation. The initial **4** crystal growth solution is depicted in the top right figure following the heating and filtering of the gel (middle).

agreement with the single crystal structure of **4** ($r_H$ as 2.1 nm). All of the results pointed to the formation of metal−organic molecular cages comparable to **4** between **1** or **2** and Pd²⁺ at 473 K. At low temperatures (below 338 K), it has been proposed that **1** or **2** and Pd²⁺ produce mixed crosslinked structures. All macrocycles changed to the most thermostable **3** upon heating at 473 K. The Pd²⁺ coordination will then be reorganized into **4** via rotating all the pyridinyl groups to the same side of the macrocycle (Fig. 8). The clear solution obtained by filtration following direct heat conversion was used to cultivate single crystals. The cell parameters of the crystal are very similar to **4**, despite the fact that the crystal quality is too inadequate to perform single-crystal X-ray diffraction analysis. So, using an in situ approach, we were able to realize the thermally sensitive nanoscale change from macrocycle to further metal−organic octahedral cage **4**. It should be noted that in this case, cage building requires a certain degree of synthetic control based just on temperature.

## Discussion

In conclusion, we present a macrocycle with three fixed atropisomers at room temperature. Here, through thermal management at various temperatures, it was possible to create the reversible or irreversible conversions between two or three rigid atropisomers. In order to create the metal−organic octahedral cage **4**, **3** coordinated with Pd²⁺. We investigated two distinct thermal-regulation pathways, namely stepwise synthesis or in situ conversion, for the **4** production using **1 or 2** and Pd²⁺. Both could control the structure of macrocyclic atropisomer molecules to further modify the self-assembly system. This work provides a complementary strategy to control the structure and properties of macrocycles, as well as related self-assembled systems.

## Methods
### Materials
Deuterated solvents were purchased from Cambridge Isotope Laboratory (Andover, MA). Other reagents were purchased commercially

(Aldrich, Acros, Adamas, Greagent, or Energy Chemical) and used without further purification.

### Synthesis of 1,5-dibromo-3-iodo-2,4-dimethylbenzene 5
A mixed solution of sulfuric acid (15.0 mL) and nitric acid (13.0 mL) was slowly dropped into 1,5-dibromo-2,4-dimethylbenzene (10.0 g, 37.9 mmol) and iodine (5.29 g, 20.8 mmol) in CH₃COOH (175 mL) at 80 °C during 1 h. The organic phase was collected with ethyl acetate (300 mL) extraction. The organic phase was washed with NaOH aqueous solution (0.01 M, 3 × 10 mL). The organic phase was dried with Na₂SO₄ and vacuum distilled. The crude product was recrystallized three times from CHCl₃, and colorless needles were obtained (10 g, 68%). ¹H NMR (400 MHz, CDCl₃, 298 K) $\delta$ (ppm): 7.76 (s, 1H), 2.67 (s, 6H) (Supplementary Fig. 1); ¹³C NMR (100 MHz, CDCl₃, 298 K) $\delta$ (ppm): 140.5, 135.5, 120.4, 108.9, 31.2 (Supplementary Fig. 2); EI HRMS ($m/z$): [M·H]⁺˙ calcd. for $C_8H_7Br_2I$, 389.8012; found, 389.7927 (Supplementary Fig. 23).

### Synthesis of 4-(3,5-dibromo-2,6-dimethylphenyl)pyridine 6
4-Pyridineboronic acid pinacol ester (10.0 g, 48.8 mmol), 1,5-dibromo-3-iodo-2,4-dimethylbenzene (**5**, 19.0 g, 48.8 mmol), Pd(PPh₃)₄ (1.69 g, 1.46 mmol), Cs₂CO₃ (79.4 g, 244 mmol), and 100 mL of toluene were added in 250 mL flask. After heating at 120 °C under argon for 24 h, the reaction mixture was filtered and the filter residue was washed with CH₂Cl₂ (50.0 mL × 3). The organic phases were combined, dried with Na₂SO₄, and vacuum distilled. The residue was purified via a silica gel (200−300 mesh) column chromatography with $n$-hexanes/ethyl acetate (20:1, $v/v$) as the eluent, which gave out product **6** as white solid (14.2 g, 85.0%). ¹H NMR (400 MHz, CDCl₃, 298 K) $\delta$ (ppm): 8.70 (d, $J = 4.00$, 2H), 7.80 (s, 1H), 7.02 (d, $J = 4.00$, 2H), 1.99 (s, 6H) (Supplementary Fig. 3); ¹³C NMR (100 MHz, CDCl₃, 298 K) $\delta$ (ppm): 150.6, 149.0, 141.7, 135.3, 134.6, 124.0, 123.2, 21.3 (Supplementary Fig. 4); ESI HRMS ($m/z$): [M + H]⁺˙ calcd. for $C_{13}H_{11}Br_2N$, 341.9311; found, 341.9312 (Supplementary Fig. 24).

### Synthesis of 4-(2,6-dimethyl-3,5-bis(4,4,5,5-tetramethyl-1,3,2-dioxaborolan-2-yl) phenyl) pyridine 7

4-(3,5-dibromo-2,6-dimethylphenyl)pyridine (**6**, 10.0 g, 29.3 mmol), bis(pinacolato)diboron (18.6 g, 73.3 mmol), Pd(dppf)$_2$Cl$_2$ (718 mg, 880 μmol), CH$_3$COOK (20.1 g, 205 mmol), and 100 mL of 1,4-dioxane were added in 250 mL flask. After heating at 105 °C under argon for 12 h, the reaction mixture was filtered and the filter residue was washed with CH$_2$Cl$_2$ (50 mL × 4). The organic phases were combined, dried with Na$_2$SO$_4$, and vacuum distilled. The residue was purified via a silica gel (200–300 mesh) column chromatography with *n*-hexanes/ethyl acetate (10:1, *v/v*) as the eluent, which gave out product **7** as white solid (9.60 g, 75.0%). $^1$H NMR (400 MHz, CDCl$_3$, 298 K) $\delta$ (ppm): 8.64 (d, *J* = 2.80, 2H), 8.11(s, 1H), 7.04 (d, *J* = 4.00, 2H), 2.15 (s, 6H), 1.33 (s, 24H) (Supplementary Fig. 5); $^{13}$C NMR (100 MHz, CDCl$_3$, 298 K) $\delta$ (ppm): 150.9, 150.2, 144.2, 143.0, 124.9, 83.6, 25.0, 20.7 (Supplementary Fig. 6); ESI HRMS (*m/z*): [M]$^{+}$ calcd. for C$_{25}$H$_{35}$B$_2$NO$_4$, 436.2834; found, 436.2831 (Supplementary Fig. 25).

### Synthesis of 1-bromo-5-iodo-2,4-dimethylbenzene 8

1,5-dibromo-2,4-dimethylbenzene (50.0 g, 189 mmol) and 400 mL ether were added into a three-neck flask under an argon atmosphere and stirred at room temperature until the solid fully dissolved. At −78.0 °C, 103 mL of *n*-BuLi (2.5 M in hexane, 258 mmol) was dropwise added in 30 min. Then adding I$_2$ (69.7 g, 275 mmol; dissolved in 70.0 mL THF) solution at room temperature during 1 h. A saturated sodium thiosulfate solution was added to quench the reaction. The aqueous phase was separated. The organic phase was collected, dried with Na$_2$SO$_4$, and vacuum distilled. The residue passes through the silica gel (200–300 mesh) flash column chromatography (*n*-hexanes as the eluent) obtain white solid **8** (56.0 g, 95.0%). $^1$H NMR (400 MHz, CDCl$_3$, 298 K) $\delta$ (ppm): 7.92(s, 1H), 7.09 (s, 1H), 2.33 (s, 3H), 2.29 (s, 3H) (Supplementary Fig. 7); $^{13}$C NMR (100 MHz, CDCl$_3$, 298 K) $\delta$ (ppm): 141.2, 140.6, 138.0, 131.7, 122.3, 97.4, 27.4, 22.5 (Supplementary Fig. 8); ESI HRMS (*m/z*): [M-H]$^{+}$ calcd. for C$_8$H$_8$BrI, 309.8927; found, 309.8845 (Supplementary Fig. 26).

### Synthesis of 4-(5,5"-dibromo-2,2",4,4',4",6'-hexamethyl-[1,1',3',1"-terphenyl]-5'-yl) pyridine 9

1-bromo-5-iodo-2,4-dimethylbenzene (**8**, 53.6 g, 172 mmol), 4-(2,6-dimethyl-3,5-bis(4,4,5,5-tetramethyl-1,3,2-dioxaborolan-2-yl)phenyl) pyridine (**7**, 30.0 g, 68.9 mmol), Pd(PPh$_3$)$_4$ (2.39 g, 2.07 mmol), K$_2$CO$_3$ (76.2 g, 552 mmol), 555 mL of toluene, 277 mL of ethanol, and 277 mL of H$_2$O were added in 2 L flask. After heating at 115 °C under argon for 12 h, the reaction was cooled to room temperature and separated. The organic phase was collected, dried with Na$_2$SO$_4$), and vacuum distilled. The residue was purified via a silica gel (200–300 mesh) column chromatography with *n*-hexanes/ethyl acetate (20:1, v/v) as the eluent, which gave out product **9** as white solid (30.0 g, 79.0%). $^1$H NMR (400 MHz, CDCl$_3$, 298 K) $\delta$ (ppm): 8.69 (d, 2H), 7.34 (s, 1H), 7.32 (s, 1H), 7.17 (t, *J* = 2.00, 2H), 7.11 (d, *J* = 3.20, 2H), 6.87 (d, *J* = 1.60, 1H), 2.38 (s, 6H), 2.05 (s, 3H), 2.01(s, 3H), 1.72 (d, *J* = 2.00, 2H) (Supplementary Fig. 9); $^{13}$C NMR (100 MHz, CDCl$_3$, 298 K) $\delta$ (ppm): 150.4, 150.3, 140.8, 140.7, 140.2, 140.1, 138.2, 138.1, 136.7, 135.4, 135.2, 133.0, 132.4, 130.3, 130.2, 124.8, 124.7, 121.7, 121.6, 22.6, 19.6, 19.5, 18.2, 18.2 (Supplementary Fig. 10); MALDI-TOF HRMS (*m/z*): [M + H]$^{+}$ calcd. for C$_{29}$H$_{27}$Br$_2$N, 550.0565; found, 550.0561 (Supplementary Fig. 27).

### Synthesis of macrocycles (1, 2, 3)

4-(5,5"-dibromo-2,2",4,4',4",6'-hexamethyl-[1,1',3',1"-terphenyl]-5'-yl) pyridine (**9**, 3.79 g, 6.89 mmol), 4-(2,6-dimethyl-3,5-bis(4,4,5,5-tetramethyl-1,3,2-dioxaborolan-2-yl)phenyl)pyridine (**7**, 3.00 g, 6.89 mmol), Pd(PPh$_3$)$_4$ (797 mg, 689 μmol), Cs$_2$CO$_3$ (15.7 g, 48.3 mmol), 1.20 L of toluene were added in 2 L flask. After heating at 120 °C under argon for 12 h, the reaction mixture was filtered and the filter residue was washed with CH$_2$Cl$_2$ (50 mL × 4). The organic phase was combined, dried with

Na$_2$SO$_4$, and vacuum distilled. The residue was purified by a silica gel (200–300 mesh) column chromatography with *n*-hexanes/acetone/methanol (10:1:0.1, *v/v/v*) as the eluent, which gives out product as white solid (yield as **1**: 126 mg, 3.20%; yield as **2**: 240 mg, yield as 6.10%; yield as **3**: 59.0 mg, 1.50%; total production and yield: 425 mg, 10.7%).

The characterization of **1**: $^1$H NMR (600 MHz, TCE-$d_2$, 298 K) $\delta$ (ppm): 8.66–8.63 (m, 6H), 8.59 (d, *J* = 5.40, 1H), 8.52 (d, *J* = 5.40, 1H), 7.27 (d, *J* = 4.20, 2H), 7.25–7.22 (m, 4H), 7.19 (d, *J* = 4.80, 1H), 7.17 (d, *J* = 5.40, 1H), 7.14 (s, 1H), 7.11 (d, *J* = 3.60, 1H), 7.07 (s, 2H), 7.06 (s, 1H), 7.05 (s, 2H), 6.98 (s, 1H), 2.22 (s, 6H), 2.19 (s, 6H), 2.15 (s, 6H), 2.12 (s, 6H), 1.85 (s, 6H), 1.78 (s, 6H), 1.72 (s, 6H), 1.69 (s, 6H) (Supplementary Fig. 11); $^{13}$C NMR (150 MHz, TCE-$d_2$, 298 K) $\delta$ (ppm): 151.2, 150.9, 150.4, 150.2, 150.0, 140.4, 139.7, 139.4, 139.3, 139.2, 138.6, 138.5, 135.0, 134.3, 133.2, 132.8, 132.6, 132.2, 132.0, 131.9, 131.6, 131.0, 130.9, 125.5, 125.3, 125.1, 20.3, 19.9, 19.7, 19.2, 19.1, 18.3, 18.1 (Supplementary Fig. 12); MALDI-TOF HRMS (*m/z*): [M]$^{+}$ calcd. for C$_{84}$H$_{76}$N$_4$, 1141.6143; found, 1141.6136 (Supplementary Fig. 28).

The characterization of **2**: $^1$H NMR (600 MHz, TCE-$d_2$, 298 K) $\delta$ (ppm): 8.65–8.62 (m, 6H), 8.57 (d, *J* = 4.80, 2H), 7.25 (d, *J* = 5.40, 2H), 7.21 (d, *J* = 4.80, 2H), 7.16 (s, 2H), 7.14–7.13 (m, 3H), 7.09 - 7.06 (m, 6H), 7.03 (s, 2H), 7.01 (s, 2H), 7.00 (s, 1H), 2.20 (s, 6H), 2.13 (d, *J* = 13.2, 18H), 1.79 (d, *J* = 7.80, 18H), 1.74 (s, 6H) (Supplementary Fig. 13); $^{13}$C NMR (150 MHz, TCE-$d_2$, 298 K) $\delta$ (ppm): 151.2, 151.0, 150.5, 150.4, 150.2, 140.0, 139.6, 139.5, 139.3, 139.15, 138.9, 138.5, 135.1, 134.9, 134.8, 133.5, 132.8, 132.2, 131.9, 131.8, 131.4, 131.3, 131.1, 130.8, 125.2, 125.1, 20.2, 20.1, 19.7, 19.6, 19.12, 18.7, 18.5, 18.4 (Supplementary Fig. 14); MALDI-TOF HRMS (*m/z*): [M]$^{+}$ calcd. for C$_{84}$H$_{76}$N$_4$, 1141.6143; found, 1141.6134 (Supplementary Fig. 29).

The characterization of **3**: $^1$H NMR (600 MHz, TCE-$d_2$, 298 K) $\delta$ (ppm): 8.60 (d, *J* = 4.80, 4H), 8.56 (d, *J* = 5.40, 4H), 7.22 (d, *J* = 4.80, 4H), 7.09–7.08 (m, 8H), 6.92 (s, 4H), 6.86 (s, 4H), 2.08 (s, 24H), 1.68 (s, 24H) (Supplementary Fig. 15); $^{13}$C NMR (150 MHz, TCE-$d_2$, 298 K) $\delta$ (ppm): 150.6, 150.2, 150.1, 139.7, 139.1, 138.9, 134.5, 131.6, 131.3, 130.5, 130.4, 124.9, 124.7, 19.6, 18.1 (Supplementary Fig. 16); MALDI-TOF HRMS (*m/z*): [M]$^{+}$ calcd. for C$_{84}$H$_{76}$N$_4$, 1141.6143; found, 1141.6135 (Supplementary Fig. 30).

### Synthesis of cage 4

In all, 4 mL tetrahydrofuran solution of **3** (20.0 mg, 17.5 μmol) were added in a 20 ml flask, then (PhCN)$_2$PdCl$_2$ (13.4 mg, 35.0 μmol in 16 mL DMF) were added and stand at 338 K for 12 h. Transparent quadrangular crystals are obtained (yield 24.5 mg, 75.0%).

## Data availability

The X-ray crystallographic data of corresponding structures reported in this study have been deposited to the Cambridge Crystallographic Data Centre (CCDC), with deposition numbers as 2280089 (**3**), 2280091 (**1**), 2280092 (**2**), or 2280095 (**4**). These data can be obtained free of charge from CCDC via www.ccdc.cam.ac.uk/data_request/cif. The new crystallographic structures of target molecules are also available within the Source data. All the other supplementary data are available from the article and its Supplementary Information files or available from the corresponding authors upon request. Source data are provided with this paper [https://doi.org/10.6084/m9.figshare.24573556][67].

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

## Acknowledgements
H.-Y. Gong is grateful to the National Natural Science Foundation of China (92156009 and 21971022), the Fundamental Research Funds for the Central Universities, the Beijing Municipal Commission of Education, and Beijing Normal University for financial Support. H.-Y. Gong also thanks the staff at BL17B1 beamline of the National Facility for Protein Science in Shanghai (NFPS), Shanghai Advanced Research Institute, CAS, also Dr. Tongling Liang and Dr. Xiang Hao in the Institute of Chemistry, Chinese Academy of Sciences, for providing technical support in X-ray diffraction data collection and analysis. X.-L. Chen. is grateful to the Science and Technology Research Project of the Hubei Provincial Department of Education(Q20222506). Support from Hubei Normal University and Hubei Key Laboratory of Pollutant Analysis and Reuse Technology is also gratefully acknowledged.

## Author contributions
J.L. synthesized the samples and carried out all the characterizations, S.L., Y.Y., Y.-J.S., J.-K.B., X.S., X.-L.C., J.C., A.-J.G., J.-F.X., and X.L. carried out partial characterization and participated in the discussion. H.W., Y.-D.Y., and H.-Y.G. supervised the work. All authors participated in the interpretation of the results and co-wrote the manuscript.

## Competing interests
The authors declare no competing interests.
