## [Peer Review File · Nature Communications]

Thermally-induced atropisomerism promotes metal-organic cage constructionReviewers' Comments:

Reviewer #1:

Remarks to the Author:

This manuscript by Liang and coworkers describes the synthesis and thermally controlled structure properties of a 4-pyridyl-substituted macrocycle and the selective formation of a Pd₁₂ cage structure by the appropriately structured isomer. As a work based on a 'molecular engineering' approach it has some merit and gives some insight into factors affecting supramolecular processes. The authors provide X-ray crystallography to characterize each isomer and also provide VT-NMR analyses establishing the interconversion or tendency to the C_{4v} at elevated temperature. The 4-pyridyl-substituted compound and its behaviour are of some interest but the authors should consider the following points in a major revision:

- 1) The authors make persistent claims about several aspects of this work fulfilling some lack in the existing literature. However, this is not always true. In particular, they claim that there are no thermally activated atropo-isomer systems (Ln 178-179) although there are several reported in the tetraphenylporphyrin system and it has been studied extensively during the last few decades (for a very recent example see Martin et al Inorg. Chem. 2021, 60(7), 5240).
- 2) Although the present case isomers can be converted to the stable isomer, this system is reminiscent of the resorcinarene systems whose isomers are generally not interconvertible. Regardless, the analogy is notable in terms of the forms of the respective molecules, i.e., chair vs. kite vs crown isomers. Also, the NMR spectra shown in Fig 4 show trends in peak structure which are remarkably similar to meso-substituted resorcinarenes.
- 3) The 'interconversion' process is quite interesting, as described in the Figures 5 and 6 and associated text. However, there seems to be some statements missing. As I understand it, isomers Cs' and Cs are metastable states but do not convert to C_{4v} in TCE at lower T. In nitrobenzene, however, the conversion to the most stable isomer is promoted and exclusive. Is there then any solvent dependence of the atropoisomer conversion or is it purely temperature-dependent? As mentioned in point (1), a comparison of this process with known atropoisomers and the rotisomerization processes would be beneficial.
- 4) Having established that the stable isomer is C_{4v}, the authors then go on to synthesize an impressive cage structure based on the well-known Pd₁₂ structural motif. What is the yield of this species?
- 5) In Figure 8, the authors show the gelatinous states of the non-C_{4v} isomers in the presence of Pd₂⁺ but there is no characterization of these states, either spectroscopic or microscopic. Is this some non-specifically-structured gel (as suggested by the polymer proposed by the authors) or is there any detectable ordering such as fibre or particle structure. Is the thermal behaviour of this gel intimately linked with the atropoisomerisation or is the gel dissolved, isomerization occurs in solution then the capsule forms? This is important for the thermal responsivity claimed by the authors.

Other matters:

- 1) I think the title of the work is misleading. It is more like: Thermally-induced atropisomerism promotes/allows metal-organic cage construction. (Or something similar)
- 2) I found Figure 3 a little unclear. Too much blue is used obscuring the N atoms. The use of the blue plane to identify the symmetry plane is also not helpful.
- 3) The authors are keen to claim that several aspects of their work are 'first' examples. These claims should be modulated either to highlight some interesting point relative to other work or just be removed. It only distracts from their description of the present study.
- 4) As mentioned above, some characterization of the gel states should be provided. Supplementary Information of this paper is comprehensive but would benefit from this data.
- 5) ORTEP and diagrams showing any significant packing motifs of the atropoisomers in their crystal structures should be given in the supplementary information.
- 6) English language should be revised throughout.

Reviewer #2:

Remarks to the Author:

The authors report a very interesting study into macrocycle synthesis, thermal interconversion and investigations into self-assembly under a range of conditions. The arguments presented are supported by a wide range of experimental approaches and the work is potentially suitable for publication in Nature Communications if the manuscript can be improved.

The paper would benefit from a very thorough proof read to pick up typographical issues, the use of incorrect words (e.g. porosity instead of pores) and other common issues. The text would benefit in certain sections from being less stop-start, greatly improving the reading experience. Other issues around units, cell parameters quoted not having esd values and almost illegible text on figures need to be addressed prior to publication. Finally, the references related to calixarenes would benefit from including the recent Atwood paper concerned with isolating spectacular nano capsules of unprecedented internal volume (<https://pubs.rsc.org/en/content/articlelanding/2023/sc/d3sc01629c>).

Reviewer #3:

Remarks to the Author:

Manuscript of Gong and co-workers describes the synthesis and characterization of three macrocyclic atropisomers, which can be interconverted into the most symmetric one by thermal treatment. In addition, these three macrocycles have been treated with Pd²⁺, leading to metal-organic complexes, two in oligomeric/polymeric structures, while the symmetric macrocycle leads to a nanocage. Notably, the authors suggest that the oligo/polymeric structures can be tune to the nanocage by thermal treatment.

The manuscript contains many mistakes, and, at the moment, I do not find some potential application of this system. In addition, considering that recently a similar macrocyclic compound has been reported (Chem. Commun., 2019, 55, 3701–3704), I believe that this manuscript lacks of the novelty required by Nature Communication. Thus, I suggest the submission in a more specialized journal.

However, some issues need to be addressed:

- abstract is not clear: in particular, it is not clear if the Pd metal ion is present in the original macrocycles or it has been added after.
- Introduction, line 65: I suggest to read some recent reports JACS 2023, 145, 2822–2829; JACS 2021, 143, 35, 14136–14146; Acc. Chem. Res. 2020, 53, 10, 2336–2346; Chem. Sci., 2020, 11, 9617–9622;
- Introduction, line 81: the authors highlight the novelty of this system also because “this is the first example of thermally responsive nanoscale transformation from macrocycle to its self-assembly”, however I suggest to read Chem. Commun., 2016, 52, 11681–11684;
- Supporting information, Figure S5: the intensity of ¹H NMR spectrum is too low. Some impurities can be present in the sample. In addition, numeration of figures in the Supporting information is not progressive (Figure S32 is repeated 2 times);
- quality of Scheme 2 must be improved. In addition, in the main text this is called Scheme, while in the caption is Figure. Reagent 4 is repeated 2 times;
- caption of Figure 4 needs to be modified: C4v-CP4 is in the middle, Cs in the bottom and Cs' is up;
- line 174: these references are not pertinent;
- Line 182: why celite or NaCl substrates have been used? in addition, in Figure S46, the temperature indicated is 473K, while in the main text 573K (line 183). Furthermore, this Figure (S46) does not show the TOTAL CONVERSION into C4v. Please clarify, also indicating the difference between right and left TLC, the solid support used.
- why nitrobenze has been used? probably for the temperature range...? please, specify.
- spectra reported in Figure S52 seem to be acquired in different solvent: in particular, the right spectrum contain nitrobenzene, while the left one not.

- Line 328: NMR titration does not indicate the formation of polymeric structures, some additional experiments should be indicated (like GPC). I suggest to change "indicate" with "suggest".
- Line 343: in Figure S51, I do not believe that the three spectra are similar. I found many differences: aromatic region (ca. 9 ppm); in the region between 6.5 and 7 ppm;
- Probably, a part of Cs and Cs' tune to C4v, and this portion is visible to NMR and DOSY measurements can be performed. However, not all the isomers tune to the cage. Can the authors support the total conversion into the cage?
- "Discussion" must be replaced with "Conclusions"

Specific changes made in response to the commands of the reviewers are as below:

Point-by-point response to referee 1:

1) The authors make persistent claims about several aspects of this work fulfilling some lack in the existing literature. However, this is not always true. In particular, they claim that there are no thermally activated atropisomers systems (Ln 178-179) although there are several reported in the tetraphenylporphyrin system and it has been studied extensively during the last few decades (for a very recent example see Martin et al Inorg. Chem. 2021, 60(7), 5240.

Response: Thanks very much for the kind remind. We scanned some literature and added to the revised manuscript as below:

“However, the complex conformer distribution and conversion especially in solution studies between more than two rigid atropisomers are still lacking.³³” is revised to “However, the complex atropisomer distribution and conversion between more than two rigid atropisomers are still lacking.^{28,61}”

And additional reference as:

61 Martin, D. J., Mercado, B. Q. & Mayer, J. M. All Four Atropisomers of Iron Tetra(o-*N,N,N*-trimethylanilinium)porphyrin in Both the Ferric and Ferrous States. Inorg. Chem. 60, 5240-5251 (2021).

2) Although the present case isomers can be converted to the stable isomer, this system is reminiscent of the resorcinarene systems whose isomers are generally not interconvertible. Regardless, the analogy is notable in terms of the forms of the respective molecules, i.e., chair vs. kite vs crown isomers. Also, the NMR spectra shown in Fig 4 show trends in peak structure which are remarkably similar to meso-substituted resorcinarenes.

Response: Appreciate so much for the kind suggestion. We compare the CP4 system with the resorcinarenes and then described their similarity in the revised manuscript as below:

It is noted that the respective CP4 isomers are analogous to meso-substituted resorcinarenes with chair (C_s' -CP4), kite (C_s -CP4), or crown (C_{4v} -CP4) structure.³⁵

And additional reference as:

35. Pineda-Castañeda, H. M., Maldonado, M. & Rivera-Monroy, Z. J. Efficient Separation of C-Tetramethylcalix[4]resorcinarene Atropisomers by Means of Reversed-Phase Solid-Phase Extraction. *ACS Omega* **8**, 231-237 (2022).

3) The 'interconversion' process is quite interesting, as described in the Figures 5 and 6 and associated text. However, there seems to be some statements missing. As I understand it, isomers C_s' and C_s are metastable states but do not convert to C_{4v} in TCE at lower T. In nitrobenzene, however, the conversion to the most stable isomer is promoted and exclusive. Is there then any solvent dependence of the atropisomer conversion or is it purely temperature-dependent? As mentioned in point (1), a comparison of this process with known atropisomers and the rotisomerization processes would be beneficial.

Response: We appreciate the reviewer for the commands.

We tested the time-dependent ^1H NMR spectra of CP4 in PhNO_2-d_5 at 393 K to determine whether there is a solvent effect in the conversion between the atropisomers of CP4. It is found that there is solvent effect shown in the conversion between the atropisomers, but the thermal dependence is still the major factor. We have described the detail in the revised manuscript and ESI as below:

In manuscript:

At 393 K, the spectra of C_s' -CP4 in PhNO_2-d_5 show gradually weakening signals and generate C_s -CP4 signals with increasing intensity ratio until the balance after 6 min (Figure 5b). Then the ratio between C_s - and C_s' -CP4 kept as 88:12. After heating C_s -CP4 solution for 1 min at 393 K, C_s' -CP4 signals appear and the system reaches dynamic equilibrium with a similar ratio between C_s - and C_s' -CP4 as 87: 13 (Figure S45, S47). It is noted that at 393 K, the reversible conversion between C_s - and C_s' -CP4 happened without C_{4v} -CP4 participation (Figure 5a-5c, Figure S44, S46). The possible

reason is the temperature is too low to induce C_s - or C_s' -CP4 passing across the energy barrier for C_{4v} -CP4 formation. The transition between C_s - and C_s' -CP4 at 393 K was regarded as a first-order reversible reaction. According to the standard equilibrium (eq.1-1), the thermal dynamics and kinetic parameters of the transition were calculated using time-dependent ^1H NMR data. These parameters include the reaction rate constants ($k_1 = (1.4 \pm 0.1) \times 10^{-2} \text{ s}^{-1}$, $k_{-1} = -(2.3 \pm 0.1) \times 10^3 \text{ s}^{-1}$), the Gibbs activated free energy ($\Delta G_{1(393 \text{ K})}^{\ddagger}$ as $111 \pm 6 \text{ KJ/M}$), the equilibrium constant (K_1 as 6.1 ± 0.2), and the Gibbs free energy ($\Delta G_{1(393 \text{ K})}^{\theta}$ as $-5.9 \pm 0.2 \text{ KJ/M}$). With the formation energy of C_s' -CP4 as standard 0, the energy diagram of the transition between C_s - and C_s' -CP4 in PhNO_2 - d_5 at 393 K can be obtained in Figure 5d

Comparing the conversion parameters from C_s' - to C_s -CP4 in PhNO_2 - d_5 or TCE - d_2 at 393 K, it is found that the reaction rate of PhNO_2 - d_5 (k_1 as $(1.4 \pm 0.1) \times 10^{-2} \text{ s}^{-1}$) is faster than that in TCE - d_2 ($k_1 = ((5.2 \pm 0.4) \times 10^{-3} \text{ s}^{-1})$). There is only similar reversible conversion process between C_s - and C_s' -CP4 (without C_{4v} -CP4 participation) in both cases. It is implied that, at the same temperature, the solvent can affect the same isomerization process, including kinetic and thermodynamic properties, but can not drive the reaction with a higher energy barrier (e.g., the conversion from C_s - or C_s' - to C_{4v} -CP4).

Figure R1 (Figure 5. in manuscript). The reversible transformation between rigid C_s - and C_s' -CP4 at 393 K. (a) Schematic representation of the transformation between C_s -, C_s' - and C_{4v} -CP4 in PhNO_2-d_5 at 393 K. (b) Time-dependent ^1H NMR spectra of C_s' -CP4 (5.3×10^{-3} M) in PhNO_2-d_5 at 393 K. (c) Time-dependent relative concentration ratio changes of C_s - (green dot) and C_s' -CP4 (blue dot) in PhNO_2-d_5 at 393 K. (d) the potential energy diagram of the conversion between C_s - and C_s' -CP4 in PhNO_2-d_5 at 393 K. The transition state is listed based on theoretical calculation.

In ESI:

Figure R2 (Figure S44. in ESI). Time-dependent ^1H NMR spectra of C_s' -CP4 (5.3×10^{-3} M) in $\text{PhNO}_2\text{-}d_5$ at 393 K (500 MHz).

Figure R3 (Figure S45. in ESI). Time-dependent ^1H NMR spectra of C_s -CP4 (5.3×10^{-3} M) in $\text{PhNO}_2\text{-}d_5$ at 393 K (500 MHz).

The conversion process in CP4 cage is compared with known atropoisomers and the rotisomerization processes in the revised manuscript as below:

In manuscript:

Compared with the reported cyclo[8](1,3-(4,6-dimethyl)benzene) (**CDMB-8**) involving two rigid atropisomers,³ the introduction of pyridine groups in **CP4** case increases the room temperature stable atropisomer kinds as three. Furthermore, as shown in the **CP4** system, lower temperature (393 K) induced reversible transformation between metastable atropisomers, and higher temperature (473 K) induced irreversible transformation from metastable forms to stable conformation. This result is not present in the **CDMB-8** system. Furthermore, temperature not less than 473 K was needed in typical rigid atropisomer conversion, including porphyrin,⁶² [4]cyclo-chrysene,¹² or [n]cyclo-4,10-pyrenylenes⁶³ etc. conversion. In **CP4** case, much lower temperature (393 K) is required for rapid conversion between the metastable atropisomer. Furthermore, compared with **CDMB-8**, there is no change on the entire molecular skeleton and the neighboring methyl groups around the flip bond (σ bond) involved in the conformational transition. It is noteworthy that the insertion of the long-range groups (i.e., pyridinyl) on **CP4** leads the difference between **CP4** and **CDMB-8**, including the type of isomer, the transition temperature, also transition process.

Compared with other macrocyclic molecules (e.g., calixarene, porphyrin, or cyclic arene) which have been extensively studied at present, chemical modifications^{15,63,64} are generally required to introduce different substituents for conformation fixing. In these systems, conformer regulation requires the participation of the guest,^{10,11,15} pH regulation,¹⁸ high temperature,⁶² etc. The example of **CP4**, which can achieve transformation control between room temperature stable conformers only through heat, complements the regulation strategy of macrocyclic structures.

And additional reference as:

62. Ishizuka, T., Tanaka, S., Uchida, S., Wei, L. & Kojima, T. Selective Convergence to Atropisomers of a Porphyrin Derivative Having Bulky Substituents at the Periphery. *J. Org. Chem.* **85**, 12856-12869 (2020).

4) Having established that the stable isomer is C_{4v} , the authors then go on to synthesize an impressive cage structure based on the well-known Pd_{12} structural motif. What is the

yield of this species?

Response: Thanks very much for the regarding. Yield and related PXRD characterization of larger-scale **CC-Pd** synthesis are given in the revised manuscript and ESI as below:

In manuscript:

Larger-scale synthesise of **CC-Pd** was carried out. **C_{4v}-CP4** (20 mg) created 24 mg **CC-Pd** sample with yield as 75% (see Figure S56). The PXRD analysis shows the pattern of larger-scale sample is similar as the simulated pattern from **CC-Pd** single crystal structure, which implies the sample structure is consistant with **CC-Pd** single crystal.

In method part:

Synthesis of Pd₁₂•(CP4)₆ (CC-Pd)

4 mL tetrahydrofuran solution of **C_{4v}-CP4** (20.0 mg, 17.5 μmol) were added in a 20 ml flask, then (PhCN)₂PdCl₂ (13.4 mg, 35.0 μmol in 16 mL DMF) were added and stand at 338 K for 12 h. Transparent quadrangular crystals are obtained (yield 24.5 mg, 75.0%).

In ESI:

Figure R4 (Figure S56. in ESI). PXRD patterns of **CC-Pd** (black line) and simulation curve obtained from single crystal data (red line), respectively.

5) In Figure 8, the authors show the gelatinous states of the non- C_{4v} isomers in the presence of Pd^{2+} but there is no characterization of these states, either spectroscopic or microscopic. Is this some non-specifically-structured gel (as suggested by the polymer proposed by the authors) or is there any detectable ordering such as fibre or particle structure. Is the thermal behaviour of this gel intimately linked with the atropisomerisation or is the gel dissolved, isomerization occurs in solution then the capsule forms? This is important for the thermal responsivity claimed by the authors.

Response: Thank you very much for your kind reminding. We added scanning electron microscopy (SEM), dynamic light scattering (DLS) and gel permeation chromatography (GPC) test and described these gel characterization results in the revised manuscript and ESI as bellow:

In manuscript:

Scanning electron microscopy (SEM) displayed all gels were formed with nanospheres or plates via. The microstructure contains a large number of micron pores (Figure S59). The dynamic light scattering (DLS) test showed that the particle sizes of C_s -CP4@2Pd or C_s' -CP4@2Pd are basically the same and much larger than the particle sizes of C_{4v} -CP4@2Pd (Figure S60). The gel permeation chromatography (GPC) results implied the average molecular weight of the polymer formed between

C_s -, or C_s' -CP4 and Pd^{2+} is about 20 times of corresponding macrocycle (Figure S61-S62). All these results suggested that C_s - or C_s' -CP4 respectively form polymeric structures with Pd^{2+} (namely C_s -CP4@2Pd or C_s' -CP4@2Pd, respectively).

In ESI:

Figure R5 (Figure S59. in ESI). SEM micrograph of C_s -CP4@2Pd (a,b), C_s' -CP4@2Pd (c,d) gels.

Figure R6 (Figure S60. in ESI). Particle size distribution of nanostructures of C_s' - (a), C_s - (b), or C_{4v} -CP4@2Pd (c) dispersed in different solvents (a₁, b₁, c₁, DMSO; a₂, b₂, c₂, PhNO₂; a₃, b₃, c₃, THF).

Figure R7 (Figure S61. in ESI). GPC spectra of C_s -CP4 (a), PhNO₂/DMSO (1/4; v/v) solution (b), C_s -CP4@2Pd (c, d is the local enlarged image).

Figure R8 (Figure S62. in ESI). GPC spectra of Cs' -CP4 (a), $PhNO_2/DMSO$ (1/4; v/v) solution (b), Cs' -CP4@2Pd (c, d is the local enlarged image).

Other matters:

1) I think the title of the work is misleading. It is more like: Thermally-induced atropisomerism promotes/allows metal-organic cage construction. (Or something similar)

Response: Thank you very much for the reviewer's comment and support. We have improved the title of the revised manuscript as below:

Thermally-induced atropisomerism promotes metal-organic cage construction

2) I found Figure 3 a little unclear. Too much blue is used obscuring the N atoms. The use of the blue plane to identify the symmetry plane is also not helpful.

Response: Thank you very much for your comment and support. We have adjusted the colors in Figure 3 in the revised manuscript. As below:

Figure R9 (Figure 3. in manuscript). Single crystal X-ray diffraction structures of CP4.

3) The authors are keen to claim that several aspects of their work are 'first' examples. These claims should be modulated either to highlight some interesting point relative to other work or just be removed. It only distracts from their description of the present study.

Response: Appreciate so much for your comment and support. We have changed "first" to "few", "unusual", or "complementary" in the revised manuscript as below:

“At present, there are also first examples of reversible/irreversible transformation between three rigid atropisomers via simple thermal regulation.” is revised as “At present, there are also few examples of reversible/irreversible transformation between three rigid atropisomers via simple thermal regulation.”

“To the best of our knowledge, this is an first example of thermally responsive

nanoscale transformation from macrocycle to self-assembly.” is revised as “To the best of our knowledge, this is an unusual example of thermally responsive nanoscale transformation from macrocycle to self-assembly.”;

“This work provides a novel strategy to control the structure and properties of macrocycles, as well as related self-assembled systems.” is revised as “This work provides a complementary strategy to control the structure and properties of macrocycles, as well as related self-assembled systems.”

4) As mentioned above, some characterization of the gel states should be provided. Supplementary Information of this paper is comprehensive but would benefit from this data.

Response: Thanks very much for the comment and support. We added SEM (Figure R5), DLS (Figure R6) and GPC (Figure R7, R8) test in the revised manuscript and ESI as shown above in the response for point 5.

5) ORTEP and diagrams showing any significant packing motifs of the atropisomers in their crystal structures should be given in the supplementary information.

Response: Thank you so much for your comments. We have improved the revised ESI with corresponding thermal ellipsoids and packing diagrams of each atropisomer in their single crystal structure as below:

Figure R10 (Figure S38. in ESI). Ellipsoid form (a) and packing structure of C_s' -CP4 along with a (b), b (c), or c (d) axis in the single crystal structure of C_s' -CP4·2CH₂Cl₂. Selected atomic distances [Å]: N(1)-N(2) 13.198(5), N(2)-N(2AA) 9.442(3), N(2) -N(3) 17.090(2), N(1)-N(3) 18.561(1).

Figure R11 (Figure S39. in ESI). Ellipsoid form (a) and packing structure of C_s -CP4 along with a (b), b (c), or c (d) axis in the single crystal structure of C_s -CP4·2CH₂Cl₂·2H₂O. Selected atomic distances [Å]: N(1)-N(2) 13.975(9), N(1)-N(3)

20.455(8), N(2)-N(3) 15.244(8), N(1)-N(4) 14.860(9).

Figure R12 (Figure S40. in ESI). Ellipsoid form (a) and packing structure of C_{4v}-CP4 along with a (b), b (c), or c (d) axis in the single crystal structure of C_{4v}-CP4·2CH₃COOC₂H₅·4CH₃CN. Selected atomic distances [Å]: N(1)-N(2) 13.443(7), N(1)-N(4) 18.795(1).

6) English language should be revised throughout.

Response: appreciate so much for your suggestions. We try our best to polish the manuscript carefully.

Point-by-point response to referee 2:

1) The paper would benefit from a very thorough proof read to pick up typographical issues, the use of incorrect words (e.g. porosity instead of pores) and other common issues. The text would benefit in certain sections from being less stop-start, greatly improving the reading experience. Other issues around units, cell parameters quoted not having esd values and almost illegible text on figures need to be addressed prior to publication. Finally, the references related to calixarenes would benefit from including the recent Atwood paper concerned with isolating spectacular nano capsules of unprecedented internal volume

(<https://pubs.rsc.org/en/content/articlelanding/2023/sc/d3sc01629c>).

Response: Thank you very much for your comments and support. We have proofread the manuscript according to the requirements of Nature Communication and corrected the wrong content. We read Atwood's paper carefully, benefited a lot, and cited the paper as below:

“The typical examples involved calixarenes,^{28,42-48} pillar[n]arene,⁴⁹ and porphyrin,^{47,50-52} etc.” is revised as “The typical examples involved calixarenes,^{24,36-42} pillar[n]arene,⁴³ and porphyrin,^{41,44-46} etc.”

And additional reference as:

42. Sikligar, K. *et al.* Nanocapsules of unprecedented internal volume seamed by calcium ions. *Chem. Sci.* **14**, 9063-9067 (2023).

49. Tuccitto, N. *et al.* The memory-driven order–disorder transition of a 3D-supramolecular architecture based on calix[5]arene and porphyrin derivatives. *Chem. Commun.* **52**, 11681-11684 (2016).

Point-by-point response to referee 3:

Manuscript of Gong and co-workers describes the synthesis and characterization of three macrocyclic atropisomers, which can be interconverted into the most symmetric one by thermal treatment. In addition, these three macrocycles have been treated with Pd²⁺, leading to metal-organic complexes, two in oligomeric/polymeric structures, while the symmetric macrocycle leads to a nanocage. Notably, the authors suggest that the oligo/polymeric structures can be tune to the nanocage by thermal treatment.

The manuscript contains many mistakes, and, at the moment, I do not find some potential application of this system. In addition, considering that recently a similar macrocyclic compound has been reported (*Chem. Commun.*, 2019, 55, 3701–3704), I believe that this manuscript lacks of the novelty required by Nature Communication. Thus, I suggest the submission in a more specialized journal.

Response: Thank you very much for your comments, it is a great help to improve our

article.

We suggested that the temperature-induced rigid conformational transformation of **CP4** has potential applications in temperature responsive materials. The assembly and conversion formed between **CP4** and Pd²⁺ could be considered as initial example.

The **CP4** case is quite different from the reported macrocyclic compound. We described it in the revised manuscript as below:

Compared with the reported cyclo[8](1,3-(4,6-dimethyl)benzene) (**CDMB-8**) with two rigid atropisomers,³ the introduction of pyridine groups in **CP4** case increases the room temperature stable atropisomer kinds as three. Furthermore, as shown in the **CP4** system, lower temperature (393 K) induced reversible transformation between metastable atropisomers, and higher temperature (473 K) induced irreversible transformation from metastable forms to stable conformation. This result is not present in the **CDMB-8** system. Furthermore, temperature not less than 473 K was needed in typical rigid atropisomer conversion, including porphyrin,⁶² [4]cyclo-chrysenes,¹² or [n]cyclo-4,10-pyrenylenes⁶³ etc. In **CP4** case, much lower temperature (393 K) is required for rapid conversion between the metastable atropisomer. Furthermore, compared with **CDMB-8**, there is no change on the entire molecular skeleton and the neighboring methyl groups around the flip bond (σ bond) involved in the conformational transition. It is noteworthy that the insertion of the long-range groups (i.e., pyridinyl) on **CP4** leads the difference between **CP4** and **CDMB-8**, including the type of isomer, the transition temperature, also transition process.

However, some issues need to be addressed:

- abstract is not clear: in particular, it is not clear if the Pd metal ion is present in the original macrocycles or it has been added after.

Response: Thank you very much for your comments, Specific experimental procedures are supplemented in the revised manuscript as below:

Abstract

“CC-Pd was generated via mixing C_{4v}-CP4 and (PhCN)₂PdCl₂ at 338 K.” is revised as “The complexation between C_{4v}-CP4 and (PhCN)₂PdCl₂ at 338 K generate metal-organic cage Pd₁₂•(CP4)₆ (i.e., CC-Pd).”

• Introduction, line 65: I suggest to read some recent reports JACS 2023, 145, 2822-2829; JACS 2021, 143, 35, 14136–14146; Acc. Chem. Res. 2020, 53, 10, 2336–2346; Chem. Sci., 2020,11, 9617-9622;

Response: Thank you very much for your comments. We have read the above article carefully and made appropriate references as below:

“It is noteworthy that ligand regulation in the metal-organic self-assembly (e.g., metal–organic cages) usually requires premodification and cumbersome synthesis, or guest addition.^{9,23,56,59-62}” is revised to “It is noteworthy that ligand regulation in the metal-organic self-assembly (e.g., metal–organic cages) usually requires premodification and cumbersome synthesis, and/or guest addition.^{7,19,49,52-59}”

And additional reference as:

56. Gu, R. & Lehn, J.-M. Constitutional Dynamic Selection at Low Reynolds Number in a Triple Dynamic System: Covalent Dynamic Adaptation Driven by Double Supramolecular Self-Assembly. *J. Am. Chem. Soc.* **143**, 14136-14146 (2021).

57. Neira, I., Blanco-Gómez, A., Quintela, J. M., García, M. D. & Peinador, C. Dissecting the “Blue Box”: Self-Assembly Strategies for the Construction of Multipurpose Polycationic Cyclophanes. *Acc. Chem. Res.* **53**, 2336-2346 (2020).

58. Jin, Y. *et al.* (Re-)Directing Oligomerization of a Single Building Block into Two Specific Dynamic Covalent Foldamers through pH. *J. Am. Chem. Soc.* **145**, 2822-2829 (2023).

59. Lin, X. *et al.* A supramolecular aggregation-based constitutional dynamic network for information processing. *Chem. Sci.* **11**, 9617-9622 (2020).

• Introduction, line 81: the authors highlight the novelty of this system also because “this is the first example of thermally responsive nanoscale transformation from

macrocycle to its self-assembly”, however I suggest to read Chem. Commun., 2016, 52, 11681–11684;

Response: Thank you very much for your comments. We have read the literature carefully and believe that the work is different with ours. In this report, the author controlled the polymer shape memory via thermal regulating the interaction between the host and guest doped on the polymer, while we regulate the assembly structure via heating induced conformation modification of the macrocycle ligand.

• Supporting information, Figure S5: the intensity of ^1H NMR spectrum is too low. Some impurities can be present in the sample. In addition, numeration of figures in the Supporting information is not progressive (Figure S32 is repeated 2 times);

Response: Thank you very much for your careful check. We improved Figure S5 and proofread the numbers of the graphs in the ESI. Related changes are shown below:

(1)

Figure R13 (Figure S5. in ESI). ^1H NMR spectrum of **3** (5.00×10^{-2} M) in CDCl_3 at 298 K (400 MHz) (red four-pointed star represents residual CHCl_3 ; orange five-pointed star represents CH_2Cl_2 , blue inverted triangle represents H_2O).

(2) **Figure S32.** Reductive elimination step from C_s' -CP4-im to C_s' -CP4 (a), from

C_s -CP4-im to C_s -CP4 (b), or from C_{4v} -CP4-im to C_{4v} -CP4 (c). The lowest formation heat values of C_s' -CP4-im, C_s -CP4-im, and C_{4v} -CP4-im in vacuum were calculated via molecular mechanics (MM+) using the force field in the HyperChem 8.0 program¹, or semiempirical methods (PM7) in MOPAC program². All the hydrogen atoms were omitted for clarity.” is revised as “**Figure S34. Reductive elimination step from C_s' -CP4-im to C_s' -CP4 (a), from C_s -CP4-im to C_s -CP4 (b), or from C_{4v} -CP4-im to C_{4v} -CP4 (c). The lowest formation heat values of C_s' -CP4-im, C_s -CP4-im, and C_{4v} -CP4-im in vacuum were calculated via molecular mechanics (MM+) using the force field in the HyperChem 8.0 program¹, or semiempirical methods (PM7) in MOPAC program². All the hydrogen atoms were omitted for clarity.”**”

• quality of Scheme 2 must be improved. In addition, in the main text this is called Scheme, while in the caption is Figure. Reagent 4 is repeated 2 times;

Response: Thank you very much for your comments. Scheme 2 has been changed to Figure 2 in the revised manuscript. The content of Figure 2 has been modified appropriately. As below:

(1) “Macrocycle CP4 was generated as shown in Scheme 2.” is revised as “**Macrocycle CP4 was generated as shown in Figure 2.**”

(2)

Figure R14 (Figure 2. in manuscript). Synthesis of macrocycle CP4. All yields refer to column chromatography purified products. More synthetic details were shown in the Methods part.

- caption of Figure 4 needs to be modified: C_{4v} -CP4 is in the middle, C_s in the bottom and C_s' is up.

Response: Thank you very much for your kind reminding. The corresponding improvement is shown below:

Figure R15 (Figure 4. in manuscript). Structures and 1H NMR spectroscopic characterizations of CP4 artopisomers.. The structure (a) and corresponding 1H NMR spectra (b) of C_s' - (up), C_{4v} - (middle), and C_s -CP4 (bottom) (each concentration as 1×10^{-2} M in TCE- d_2 at 298 K (500 MHz)).

- line 174: these references are not pertinent;

Response: Thank you very much for your kind reminding. We have made adjustments

for the references here. as below:

“Biaryl atropisomerism has ubiquity in natural products.^{4,6,33}” is revised as “Biaryl atropisomerism has ubiquity in natural products.”

• Line 182: why celite or NaCl substrates have been used? in addition, in Figure S46, the temperature indicated is 473 K, while in the main text 573 K (line 183). Furthermore, this Figure (S46) does not show the TOTAL CONVERSION into C_{4v} . Please clarify, also indicating the difference between right and left TLC, the solid support used.

Response: Thank you very much for your kind reminding.

(1) If no NaCl substrates are used, CP4 will be fully carbonized at 573K.

(2) The 473 K given in the caption of the original Figure S36 is wrong, we have changed it to 573 K. It has been clarified that C_s -, C_s' -CP4 were quantitatively converted to C_{4v} -CP4 at 573K. In the revised manuscript, we clarify the difference between the left and right TLC in Figure S41, and solid support is used. As below:

“Figure R16 (before Figure S36., now Figure S41 in ESI). Conversion from C_s' -CP4 (left) or C_s -CP4 (right) to C_{4v} -CP4 in solid state with NaCl substrate at 573 K. Eluent was *n*-hexane/acetone/methanol (left, 10:1:0.1, right, 10:1:0.05; v/v/v).”

• why nitrobenze has been used? probably for the temperature range...? please, specify.

Response: Thank you very much for your notice. We have explained why in the revised manuscript as below:

The temperature-dependent ^1H NMR spectra of C_s -, C_s' -, or C_{4v} -CP4 were recorded in

PhNO₂-d₅ from 298 to 473 K since PhNO₂-d₅ is commercialized deuterated solvent with the highest boiling point (483 K)

- spectra reported in Figure S52 seem to be acquired in different solvent: in particular, the right spectrum contain nitrobenzene, while the left one not.

Response: Thank you very much for your notice. We made an error in processing the spectrum, which has been corrected in the revised manuscript. As below:

“Figure R17 (before Figure S58., now Figure S65. in ESI). ¹H DOSY NMR spectra of ligand C_{4v}-CP4 (a) and C_{4v}-CP4@2Pd (b) (600 MHz, THF-d₈/DMSO-d₆ (1/4; v/v), 298 K).”

- Line 328: NMR titration does not indicate the formation of polymeric structures, some additional experiments should be indicated (like GPC). I suggest to change "indicate" with "suggest".

Response: Thank you very much for your comment.

(1) We added SEM (Figure R5), DLS (Figure R6) and GPC (Figure R7, R8) test in the revised manuscript and ESI as below:

In manuscript:

Scanning electron microscopy (SEM) displayed all gels were formed with nanospheres or plates via. The microstructure contains a large number of micron pores (Figure S59). The dynamic light scattering (DLS) test showed that the particle sizes of

C_s -CP4@2Pd or C_s' -CP4@2Pd are basically the same and much larger than the particle sizes of C_{4v} -CP4@2Pd (Figure S60). The gel permeation chromatography (GPC) results implied the average molecular weight of the polymer formed between C_s -, or C_s' -CP4 and Pd²⁺ is about 20 times of corresponding macrocycle (Figure S61-S62). All these results suggested that C_s - or C_s' -CP4 respectively form polymeric structures with Pd²⁺ (namely C_s -CP4@2Pd or C_s' -CP4@2Pd, respectively).

(2) We have changed "indicate" with "suggest" as below:

“The results indicate that C_s - or C_s' -CP4 respectively form polymeric structures with Pd²⁺ (namely C_s -CP4@2Pd or C_s' -CP4@2Pd, respectively).” is revised to “All these results suggested that C_s - or C_s' -CP4 respectively form polymeric structures with Pd²⁺ (namely C_s -CP4@2Pd or C_s' -CP4@2Pd, respectively).”

• Line 343: in Figure S51, I do not believe that the three spectra are similar. I found many differences: aromatic region (ca. 9 ppm); in the region between 6.5 and 7 ppm;

Response: Thank you very much for your comments. The differences in the contents of Figure S51 have been explained in the modified ESI as below:

It is suggested that the difference between C_s -, C_s' -, and C_{4v} -CP4@3Pd is due to the concentration difference. C_{4v} -CP4@3Pd created precipitate in the initial assembly stage, while C_s - or C_s' -CP4@3Pd do not generate a large amount of precipitate during the heating promoted CC-Pd conversion. It is supposed that CC-Pd concentrations are higher in the latter two cases.

• Probably, a part of C_s and C_s' tune to C_{4v} , and this portion is visible to NMR and DOSY measurements can be performed. However, not all the isomers tune to the cage. Can the authors support the total conversion into the cage?

Response: Thank you very much for your comments. C_s -, C_s' -CP4@3Pd will produce a large number of black carbides in PhNO₂-d₅ at high temperature (473 K). NMR and DOSY measurements did not detect the signal of isomers, but it does not rule out the possibility of the presence of isomers in black substances.

- "Discussion" must be replaced with "Conclusions"

Response: Thank you very much for your kind reminding. We checked the Nature Communications template again and found that "Discussion" was used in the template.

Reviewers' Comments:

Reviewer #1:

Remarks to the Author:

The manuscript by Liang and coworkers has been revised and improved substantially and, in my opinion, is suitable for publication if the authors consider the following points:

- 1) The authors persist with the view that atropisomerism involving two or more isomers is unique to their system, which it is not. I have already directed them to the most obvious other contrasting but relevant type of system. They only need to reduce the claim of novelty with respect to this phenomenon and update using relevant citations.
- 2) Figure 3 remains difficult to understand. Please improve.
- 3) As mentioned by another reviewer, there remains a question of novelty based on the known macrocyclic compound, which might also apply to the PdL type cages also utilized by the authors. In this case, the authors have demonstrated a certain level of synthetic control necessary to reach the cage structure, which could be noteworthy.

I have been asked to check the revisions made according to the comments of Reviewer #2.

Basically, the authors appear to have added items like esd values for bond lengths and other units requested by the reviewer #2. Where possible, the authors should add esd's for torsion angles mentioned in their discussion of the macrocycles starting at Ln 114 of the revised manuscript. There are some strange additions such as some double subscripts of some parameters (e.g., in the paragraph starting at Ln 253), which should be revised.

Reviewer #2 specifically requested the proper revision of the English language of the manuscript, which the authors have not done. The authors should consult a service in order to make the English language acceptable. The manuscript would certainly benefit from this.

Reviewer #3:

Remarks to the Author:

The authors addressed the issue raised during the first step.

Specific changes made in response to the commands of the reviewers are as below:

Point-by-point response to referee 1:

1) The authors persist with the view that atropisomerism involving two or more isomers is unique to their system, which it is not. I have already directed them to the most obvious other contrasting but relevant type of system. They only need to reduce the claim of novelty with respect to this phenomenon and update using relevant citations.

Response: Appreciate so much for your comment and support. We have removed "unique" from the revised manuscript as below:

“In addition to offering a unique thermally accelerated method for modifying self-assembled systems using macrocyclic building blocks, this study also has the potential to develop the "nanoscale transformation" material with a thermal response.” is revised as “In addition to offering a thermally accelerated method for modifying self-assembled systems using macrocyclic building blocks, this study also has the potential to develop the "nanoscale transformation" material with a thermal response.”

“Here, through thermal management at various temperatures, it was possible to create the unique reversible or irreversible conversions between two or three rigid atropisomers.” is revised as “Here, through thermal management at various temperatures, it was possible to create the reversible or irreversible conversions between two or three rigid atropisomers.”

2) Figure 3 remains difficult to understand. Please improve.

Response: Thank you very much for the reviewer’s comment and support. We have improved Figure 3 of the revised manuscript as below:

Figure R1 (Fig 3. in manuscript). Single crystal X-ray diffraction structures of CP4.

3) As mentioned by another reviewer, there remains a question of novelty based on the known macrocyclic compound, which might also apply to the PdL type cages also utilized by the authors. In this case, the authors have demonstrated a certain level of synthetic control necessary to reach the cage structure, which could be noteworthy.

Response: Thank you very much for the reviewer's support. As the suggestion, we made the improvement in the revised manuscript as below:

It should be noted that in this case, cage building requires a certain degree of synthetic control based just on temperature.

Basically, the authors appear to have added items like esd values for bond lengths and other units requested by the reviewer #2. Where possible, the authors should add esd's for torsion angles mentioned in their discussion of the macrocycles starting at Ln 114 of the revised manuscript. There are some strange additions such as some double subscripts of some parameters (e.g., in the paragraph starting at Ln 253), which should be revised.

Response: We appreciate the reviewer for the commands.

a. We have added esd's for torsion angles in the revised manuscript as below:

The torsion angles between neighboring **B** and **P** range from 117.1(1)^o to 190.7(5)^o (Supplementary Fig. 36).

The torsion angles between the adjacent aromatics are between 117.0(2)^o and 119.2(2)^o (Supplementary Fig. 38).

The four **B** units can form another pyramid when they all point to a different side of the macrocycle. In this case, the torsion angle between the adjacent planes is between 117.5(4)^o and 121.2(8)^o degrees (Supplementary Fig. 40).

b. We revised the parameters with double subscripts in the revised manuscript as below:

the Gibbs activated free energy values ($\Delta G_{1(473\text{ K})}^{\ddagger}$ as $128 \pm 6\text{ kJ mol}^{-1}$; $\Delta G_{2(473\text{ K})}^{\ddagger}$ as $142 \pm 7\text{ kJ mol}^{-1}$) (Supplementary Table 4), the equilibrium constants (K_1 as 6.1 ± 0.2 ; K_2 as 92 ± 5), and the Gibbs free energy values ($\Delta G_{1(473\text{ K})}^{\theta}$ as $-4.3 \pm 0.3\text{ kJ mol}^{-1}$; $\Delta G_{2(473\text{ K})}^{\theta}$ as $-13 \pm 1\text{ kJ mol}^{-1}$) (Supplementary Table 4).

Reviewer #2 specifically requested the proper revision of the English language of the manuscript, which the authors have not done. The authors should consult a service in order to make the English language acceptable. The manuscript would certainly benefit from this.

Response: Thank you very much for your kind reminding. We make all attempts to improve the manuscript's English with the assistance of a native speaker.

Point-by-point response to referee 3:

The authors addressed the issue raised during the first step.

Thank you very much for the reviewer's support.

Reviewers' Comments:

Reviewer #1:

Remarks to the Author:

The authors have revised the manuscript according to the comments of the reviewers.